# The CLIP-domain serine protease CLIPC9 regulates melanization downstream of SPCLIP1, CLIPA8, and CLIPA28 in the malaria vector *Anopheles gambiae*

**Gregory L. Sousa**[1], **Ritika Bishnoi**[2], **Richard H. G. Baxter**[2], **Michael Povelones**[1]*

**1** Department of Pathobiology, School of Veterinary Medicine, University of Pennsylvania, Philadelphia, Pennsylvania, United States of America, **2** Department of Medical Genetics and Molecular Biochemistry, Lewis Katz School of Medicine, Temple University, Philadelphia, Pennsylvania, United States of America

* mpove@vet.upenn.edu

**Data Availability Statement:** All relevant data are within the manuscript and its Supporting Information files.

## Abstract

The arthropod melanization immune response is activated by extracellular protease cascades predominantly comprised of CLIP-domain serine proteases (CLIP-SPs) and serine protease homologs (CLIP-SPHs). In the malaria vector, *Anopheles gambiae*, the CLIP-SPHs SPCLIP1, CLIPA8, and CLIPA28 form the core of a hierarchical cascade downstream of mosquito complement that is required for microbial melanization. However, our understanding of the regulatory relationship of the CLIP-SPH cascade with the catalytic CLIP-SPs driving melanization is incomplete. Here, we report on the development of a novel screen to identify melanization pathway components based on the quantitation of melanotic mosquito excreta, eliminating the need for microdissections or hemolymph enzymatic assays. Using this screen, we identified CLIPC9 and subsequent functional analyses established that this protease is essential for the melanization of both *Escherichia coli* and the rodent malaria parasite *Plasmodium berghei*. Mechanistically, septic infection with *E. coli* promotes CLIPC9 cleavage and both full-length and cleaved CLIPC9 localize to this bacterium in a CLIPA8-dependent manner. The steady state level of CLIPC9 in the hemolymph is regulated by thioester-containing protein 1 (TEP1), suggesting it functions downstream of mosquito complement. In support, CLIPC9 cleavage is inhibited following SPCLIP1, CLIPA8, and CLIPA28 knockdown positioning it downstream of the CLIP-SPH cascade. Moreover, like CLIPA8 and CLIPA28, CLIPC9 processing is negatively regulated by serine protease inhibitor 2 (SRPN2). This report demonstrates how our novel excretion-based approach can be utilized to dissect the complex protease networks regulating mosquito melanization. Collectively, our findings establish that CLIPC9 is required for microbial melanization in *An. gambiae* and shed light on how the CLIP-SPH cascade regulates this potent immune response.

**Funding:** This work was supported by an National Institute of Allergy and Infectious Diseases (https://www.niaid.nih.gov/) grant R01AI139060 of the National Institutes of Health to MP, a University of Pennsylvania MSTP Program T32 grant GM007170 for GLS, a VMD-PhD Program T32 grant AI070077 for GLS, a parasitology T32 grant AI007532 for GLS, and a National Institute of General Medical Sciences (https://www.nigms.nih.gov/) grant R01GM114358 of the National Institutes of Health to RHGB. The funders had no role in study design, data collection and analysis, decision to publish, or preparation of the manuscript.

**Competing interests:** The authors have declared that no competing interests exist.

## Author summary

Mosquito vector competence for *Plasmodium*, antifungal defense, and lifespan are all influenced by the melanization response. Despite its importance, our understanding of the proteins comprising the *An. gambiae* melanization cascade is incomplete. To streamline the discovery of melanization pathway components in this disease vector, we developed a screening method that is able to identify proteins with far fewer mosquitoes than other approaches. This technique facilitated our discovery that the serine protease CLIPC9 is required for the melanization of *E. coli* bacteria and malaria parasites. CLIPC9 activation and localization to bacteria is regulated by members of the CLIPA subfamily, highlighting that these serine protease homologs broadly regulate the melanization immune response. Traditionally viewed as 'co-factors' for the prophenoloxidase activating CLIPBs, this work demonstrates that CLIPAs can also control the activation of the catalytic proteases driving melanization. This work identifies a new player in the melanization hierarchy and provides the field with an efficient approach for dissecting the complex protease cascades controlling this important immune response.

## Introduction

Malaria is the most lethal vector-borne disease and kills over 400,000 individuals each year [1]. *Plasmodium* parasites cause malaria and are predominantly transmitted by *An. gambiae* mosquitoes during blood feeding. The parasite life cycle in the mosquito begins when *Plasmodium* gametocytes enter the midgut lumen with the blood meal and fuse to form invasive ookinetes that traverse the midgut epithelium [2]. *An. gambiae* mounts a potent, complement-like immune response (hereafter referred to as mosquito complement) that eliminates the majority of ookinetes that reach the hemolymph-imbued midgut basal lamina by either lysis or melanization [3–5].

Melanization is an arthropod-specific reaction involving the production of melanin on microbial surfaces or wounds [6]. Phenoloxidase (PO) is an essential melanization enzyme that is activated by cleavage of its pro-form (PPO) during the terminal step of a complex protease cascade composed largely of CLIP-SPs from subfamilies B and C and CLIP-SPHs from subfamily A (collectively referred to as CLIPs) [7, 8]. An evolutionarily conserved cascade hierarchy governing PPO activation has been revealed by biochemical analyses in hemolymph-laden insects such as *Manduca sexta* and genetic investigations using *Drosophila melanogaster* [9]. Cascade components are sequentially activated through specific proteolytic cleavage events initiated by the autoactivation of a modular serine protease (Mod-SP) whose diverse domains link microbial pattern recognition and downstream zymogen activation [10–16]. Active Mod-SP cleaves a CLIPC, which then targets a terminal CLIPB protease [17–21]. CLIPBs are PPO activating proteases (PAPs) that cleave PPO into active PO often in coordination with catalytically inactive CLIPA co-factors [14, 22–32]. In addition to melanin entrapment, noxious melanogenic byproducts such as quinones and reactive oxygen intermediates are implicated in killing microbes that contact the hemocoel [33]. Serine protease inhibitors of the serpin superfamily protect the host from excessive generation of these toxic intermediates by eliminating active proteases via suicide inhibition [27, 34–43]. The elucidation of this core melanization hierarchy has informed investigations of the *An. gambiae* melanization cascade, however considerable knowledge gaps remain, particularly with respect to the relationships between the positive regulatory CLIPAs and the catalytic CLIP-SPs.

Our mechanistic understanding of the CLIPAs positively regulating *An. gambiae* melanization is largely confined to their relationships with TEP1 (the mosquito complement

opsonin) and the serine protease inhibitor, SRPN2. TEP1 circulates in the hemolymph as a full-length (TEP1-Full) and a cleaved (TEP1$_{cut}$) form with a reactive thioester motif that binds microbial surfaces [44]. TEP1$_{cut}$ opsonization drives a convertase mechanism that amplifies TEP1$_{cut}$ accumulation from circulating TEP1-Full and ultimately promotes microbial melanization [4, 45–50]. SPCLIP1 (also called CLIPA30) localizes to microbial surfaces in a TEP1-dependent manner and positively regulates melanization by promoting TEP1 convertase activity [46, 51]. SPCLIP1 is also required for the cleavage-induced activation of CLIPA8 and CLIPA28, which collectively form the core of a sequentially activated CLIP-SPH cascade essential for melanization [46, 52, 53]. The cleavages of CLIPA8 and CLIPA28 are inhibited by SRPN2 and silencing this negative regulator results in the spontaneous melanization of self-tissue [40, 53, 54].

The hierarchical relationships of SPCLIP1, CLIPA8, and CLIPA28 suggest interactions with multiple catalytic proteases driving melanization [53]. Evidence from other insect models indicates that CLIPAs are cleaved by CLIPBs [14, 24, 28, 31, 32, 55–57], however, silencing several of the putative PAPs with strong melanization phenotypes such as CLIPB4, CLIPB8, CLIPB9, CLIPB14, and CLIPB17 [5, 42, 58, 59] did not effect CLIPA28 activation [53]. This suggests that some positive regulatory CLIPAs may be activated by and/or regulate upstream proteases. Despite numerous examples of penultimate CLIPCs activating CLIPBs in other insects [14, 17–21], none are linked to melanization in *An. gambiae*. The identification and characterization of CLIPCs required for *An. gambiae* melanization is essential for understanding how the CLIP-SPH cascade regulates this immune response. Here, we report the development and use of a simple screen for dissecting the *An. gambiae* melanization cascade. This technique facilitated our identification of CLIPC9 as a protease required for microbial melanization and the characterization of its biochemical relationship with TEP1, the positive regulatory CLIP-SPH cascade, and SRPN2. This report describes the first CLIPC required for *An. gambiae* melanization and illustrates the extensive regulatory control of the CLIPA subfamily over this immune response.

## Results

### Quantitation of melanotic excreta is a streamlined method to identify proteins required for the melanization immune response

The melanization immune response is typically quantified in mosquitoes using PO enzyme activity assays or determining the extent of microbial melanotic encapsulation *in situ* [54, 60]. These assays are challenging due to the mosquito's small size and limited volume of extractable hemolymph. However, we made the observation that *An. gambiae* septically infected with *E. coli* excrete brown material onto the filter papers lining the bottom of their housing cups (Fig 1A), and we hypothesized these excreta were positively associated with microbial melanization and could therefore be utilized as a screen to identify components required for this response. To test these possibilities, we developed an image processing and analysis pipeline (S1A Fig) to quantify the total area of absorbed spots on a filter paper (hereafter referred to as total spot area) and assessed its ability to discriminate between mosquitoes injected with PBS and *E. coli*. Filter papers collected 12 h post-injection could be blindly sorted (S1B Fig), and image analysis revealed that the total spot area was 18.4 times greater after *E. coli* challenge relative to PBS injection (Fig 1B). We then used RNAi-mediated gene silencing to validate this method, which we now refer to as the Melanization-associated Spot Assay (MelASA), by comparing the total spot area between *E. coli* challenged CLIPA8 knockdown and ds*LacZ*-treated (control) mosquitoes. We reasoned that if the induced excreta were coupled with systemic melanization, then silencing this essential regulator would reduce both PO enzymatic activity and total spot

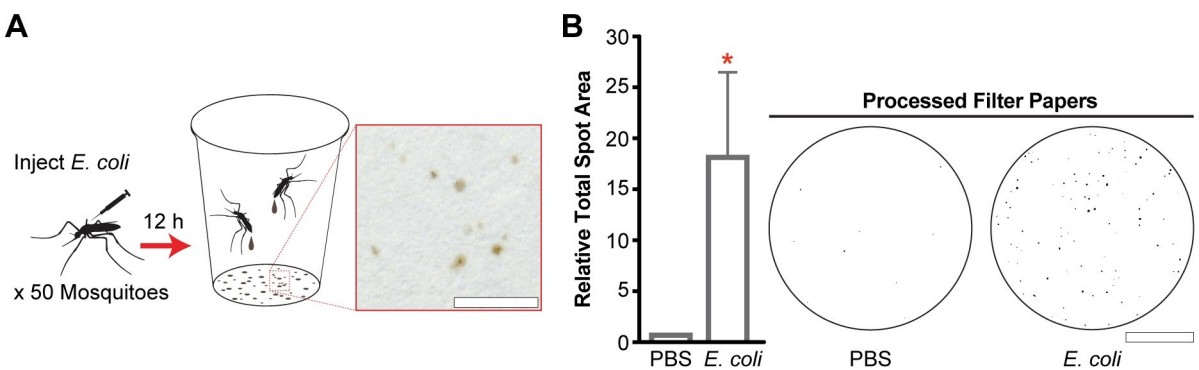

**Fig 1. Mosquitoes excrete brown material after *E. coli* septic infection.** **(A)** Schematic diagram and representative example of *E. coli* septic infection-induced excreta 12 h post-challenge. Inset image scale bar is 5 mm. **(B)** The total spot area from 50 mosquitoes 12 h after PBS or *E. coli* challenge (n = 4; one sample t-test). Processed filter paper images are from a representative replicate. Scale bar is 2.5 cm. Error bars are ± SD. Asterisks denote statistical significance ($^*$p ≤ 0.05). All experimental replicates are from independent mosquito generations.

area (Fig 2A). Indeed, 4 h after *E. coli* injection CLIPA8 knockdowns have reduced PO activity (Fig 2B), as previously reported [54], and 4.2-fold reduced total spot area (Fig 2C) relative to controls. The total spot area was also 4.8-fold lower in CLIPA8 knockdowns 12 h after *E. coli* injection (Fig 2D). The 12 h timepoint was selected for subsequent experiments due to the strength of the MelASA phenotype and lack of *E. coli* challenge-associated mosquito mortality. Collectively, these results show that total spot area is functionally linked with *E. coli* melanization, and that MelASA is a non-destructive screen capable of identifying melanization cascade components without hemolymph extractions, enzymatic assays, or microdissections.

While each MelASA used 50 mosquitoes (hereafter referred to as Standard MelASA), we sought to reduce this number to facilitate the screening of larger numbers of melanization pathway candidates. Similar to the Standard MelASA, downsizing to 20 mosquitoes housed in a smaller cup resulted in a 4.8-fold total spot area reduction in CLIPA8 knockdowns compared to controls (Fig 2E). The miniaturized MelASA (Mini-MelASA) affords the option to screen candidates with a fraction of the required animals and can be used interchangeably with Standard MelASA. In support, silencing SPCLIP1 (Fig 2F) and CLIPA28 (Fig 2G) reduced total spot area comparably to CLIPA8 knockdown. The Standard and Mini-MelASAs will collectively be referred to as MelASA. These results demonstrate that MelASA is a streamlined alternative to quantify the melanization immune response (Fig 2H).

## MelASA identifies CLIPC9 as a protease required for microbial melanization

Following the establishment of MelASA as a tool to identify proteins required for the melanization immune response, we turned our attention to the uncharacterized *An. gambiae* CLIPC subfamily. The conserved hierarchical relationships governing PPO activation in many insects [9] led us to reason that one or more CLIPCs may be required for melanization. We produced dsRNA against 8 CLIPCs and advanced 7 candidates for MelASA with average silencing efficiencies of 49–83% (S2A Fig). CLIPC2 was excluded due to poor silencing efficiency, and we were unable to generate dsRNA against CLIPC5. Following the inception of this project, 3 additional CLIPCs were identified (CLIPC12-14) which were not included in this screen and CLIPC7 was reclassified as a CLIPE [51]. Notably, CLIPC9 was the only screened candidate with a significant effect on MelASA, as ds*CLIPC9* treatment reduced total spot area 3.7-fold 12 h after *E. coli* challenge relative to control (Fig 3A). Indeed, the total spot area reduction and

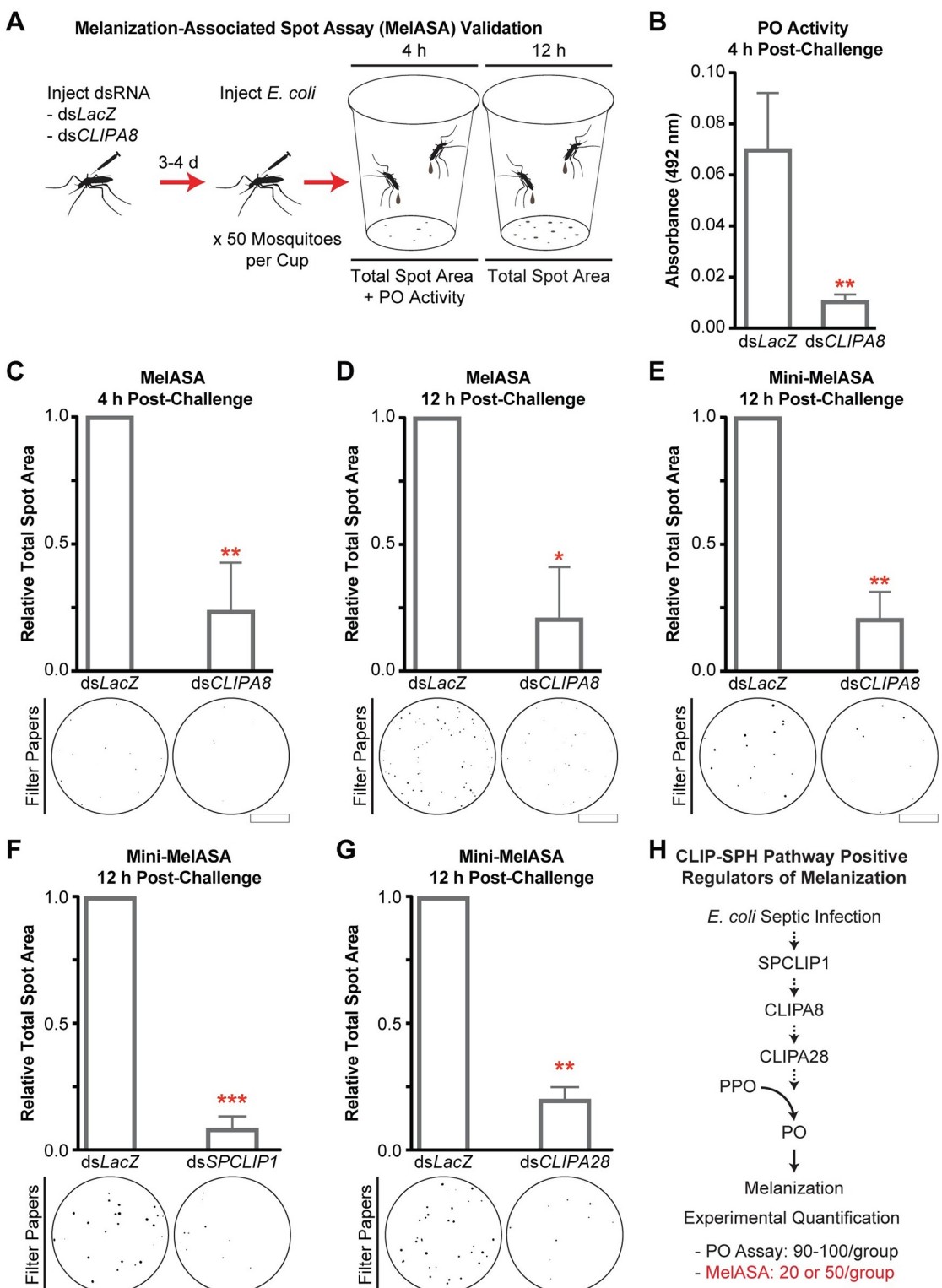

**Fig 2. MelASA links PO activity with quantifiable melanotic excreta. (A)** Following gene silencing, 50 mosquitoes per treatment group were injected with *E. coli* and placed into cups to capture excretions. Hemolymph PO enzyme activity was measured 4 h post-challenge and total spot area was assessed at both 4 and 12 h. **(B)** Hemolymph PO enzyme activity, measured as absorbance at 492 nm, 4 h after *E. coli* challenge (n = 3; unpaired t-test). Total spot area from 50 mosquitoes **(C)** 4 h after *E. coli* challenge (n = 4; one sample t-test) and **(D)** 12 h after *E. coli* challenge (n = 3; one sample t-test). Scale bars are 2.5 cm. Mini-MelASA total spot area

from 20 mosquitoes following (**E**) CLIPA8, (**F**) SPCLIP1, and (**G**) CLIPA28 knockdown 12 h post-*E. coli* injection (n = 3 for each; one sample t-test). Scale bars are 1.3 cm. (**H**) Pathway diagram of CLIP-SPHs successfully identified via Standard or Mini-MelASA after *E. coli* septic infection. Dashed arrows represent uncharacterized enzymatic steps. All error bars are ± SD. Asterisks denote statistical significance (*p ≤ 0.05, **p ≤ 0.01, ***p ≤ 0.001). All experimental replicates are from independent mosquito generations.

filter paper appearance were comparable to CLIPA8 knockdown (Fig 3B). CLIPC9 silencing was specific and did not reduce the expression of the other candidates or the CLIP-SPH pathway members (S2B Fig). In addition, ds*CLIPA8*, ds*SPCLIP1*, and ds*CLIPA28* treatments did not reduce CLIPC9 expression (S2C Fig). Collectively, these results suggest the observed Mel-ASA phenotypes are specific to the intended targeted genes and strongly support an essential role for CLIPC9 in the melanization immune response.

## CLIPC9 is required for *E. coli* melanization

To confirm the MelASA identification of CLIPC9 as a protease required for microbial melanization, we next measured PO enzyme activity after *E. coli* septic infection in CLIPC9 knockdowns. In contrast to the 5.8-fold increase in PO activity 4 h after *E. coli* infection in ds*LacZ*-treated mosquitoes relative to PBS injection, there was no inducible PO activity in CLIPC9 knockdowns relative to PBS challenged controls (Fig 4A). This result suggests that *E. coli* melanization requires CLIPC9.

Bacterial septic infection results in microbial melanization along the periostial regions of the mosquito heart and surrounding tissue [48, 54, 60]. To further validate our findings, we categorically scored this phenomenon 2 d after *E. coli* injection in ds*CLIPC9*-treated mosquitoes. CLIPA8 knockdowns were included as a positive control for blocking melanization [54]. *E. coli* septic infection resulted in most of the ds*LacZ*-treated mosquitoes being scored in the moderate and severe melanization categories, while the majority of CLIPC9 knockdowns lacked melanized bacteria. Indeed, the prevalence of mosquitoes with no observable melanization was significantly increased in CLIPC9 knockdowns and indistinguishable from the ds*CLIPA8*-treated group (Fig 4B). CLIPC9 knockdown did not compromise mosquito survival following *E. coli* septic infection (S3 Fig), which is similar to a previous report for CLIPA8

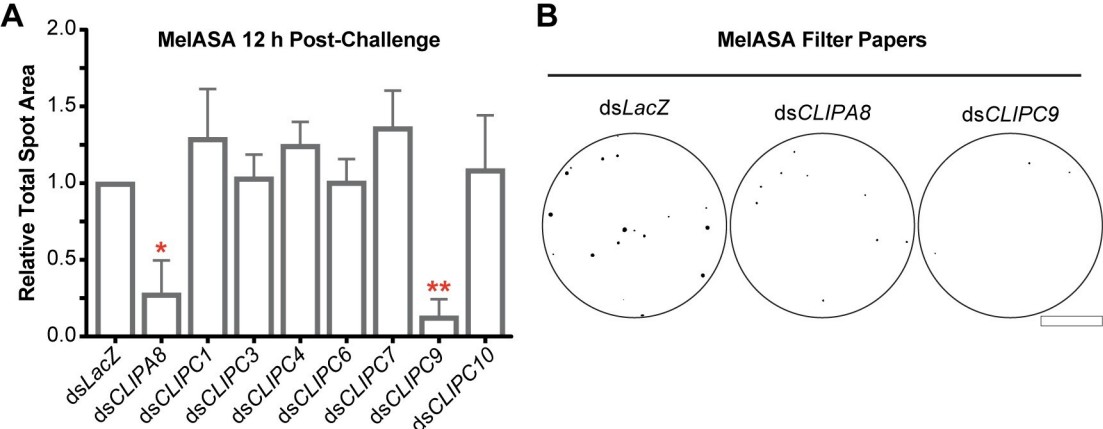

**Fig 3. MelASA links CLIPC9 to the melanization immune response. (A)** MelASA analysis of screened candidates 12 h post-*E. coli* injection. Data are compiled from one Standard and two Mini-MelASAs. Mean relative total spot areas were compared to control with one sample t-tests. Error bars are ± SD. (**B**) Representative Mini-MelASA filter papers from control, CLIPA8, and CLIPC9 knockdowns. Scale bar is 1.3 cm. Asterisks denote statistical significance (*p ≤ 0.05, **p ≤ 0.01). Data are from independent biological replicates.

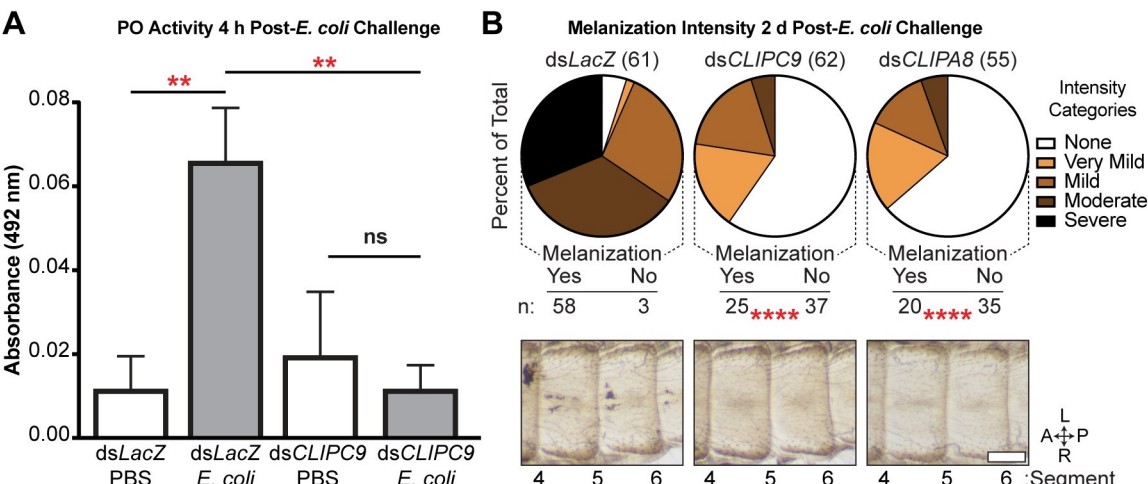

**Fig 4. CLIPC9 silencing blocks *E. coli* melanization. (A)** Hemolymph PO activity 4 h after *E. coli* or PBS injection in ds*LacZ*- and ds*CLIPC9*-treated mosquitoes. Error bars are ± SD. Means were compared with a one-way ANOVA and Tukey's multiple comparisons test. **(B)** Melanization intensity scores and representative cuticle photomicrographs 2 d post-*E. coli* injection in ds*LacZ*-, ds*CLIPC9*-, and ds*CLIPA8*-treated mosquitoes. Pie charts indicate percent of total observations blindly scored for each intensity category. Total treatment group sizes are in parentheses. The ds*CLIPC9*- and ds*CLIPA8*-treatment groups were compared to ds*LacZ*-treated control using Fisher's exact test. Images span abdominal segments 4–6 with mosquitoes oriented along the anterior-posterior (A-P) and left-right (L-R) axes as indicated. The scale bar is 250 μm. Asterisks denote statistical significance (\**p ≤ 0.01, \*\*\*\*p ≤ 0.0001). All data are pooled from three independent biological replicates.

[54]. Collectively, our findings support an essential role for CLIPC9 in *E. coli* melanization and suggest that CLIPC9 may participate in a core melanization pathway with CLIPA8.

## *P. berghei* ookinete melanization is inhibited by CLIPC9 silencing

*Plasmodium* ookinete melanization in the *An. gambiae* G3 strain is inhibited by the C-type lectins CTL4 and CTLMA2, which are present in the hemolymph as a heterodimer [5, 61–63]. Silencing either CTL4 or CTLMA2 enhances the melanization-induced killing of *P. berghei* ookinetes leading to a reduction in viable oocysts [5, 62]. Ookinete killing in the CTL4/CTLMA2 knockdown refractory background requires melanization, as co-silencing of CTL4 and either CLIPA8 or CLIPA28 blocks this response and rescues viable parasites [5, 53]. Similar to what we and others [5] observed for CLIPA8 single knockdowns, CLIPC9 silencing does not alter either the prevalence or intensity of viable oocysts or melanized ookinetes (Fig 5A) allowing us to specifically assess its role in CTL4/CTLMA2 knockdown-induced melanization. Notably, co-silencing of CLIPC9 and CTL4 significantly reduced ookinete melanization (Fig 5B and 5C) compared to CTL4 knockdown alone, however this reduction was not coincident with a rescue of viable oocysts (Fig 5D). In contrast, and consistent with a previous report [5], the reduction in ookinete melanization following CLIPA8 and CTL4 co-silencing (Fig 5B and 5C) was concurrent with an oocyst rescue (Fig 5D). We note that the reduction in ookinete melanization was slightly more potent in the ds*CTL4*/ds*CLIPA8* treatment group as it reduced the median number of melanized ookinetes back to the level observed in control. Thus, while these data show that CLIPC9 is essential for *P. berghei* ookinete melanization, they also indicate that parasite viability remains compromised in ds*CLIPC9*/ds*CTL4*-treated mosquitoes.

While performing these experiments we found that filter papers removed from cups housing naïve, ds*CTL4*-treated mosquitoes had 3.7-fold increased total spot area compared to control (S4A Fig), prompting us to examine these mosquitoes for evidence of melanization. Indeed, we found prominent thoracic melanization in 32.8% of ds*CTL4*-treated mosquitoes

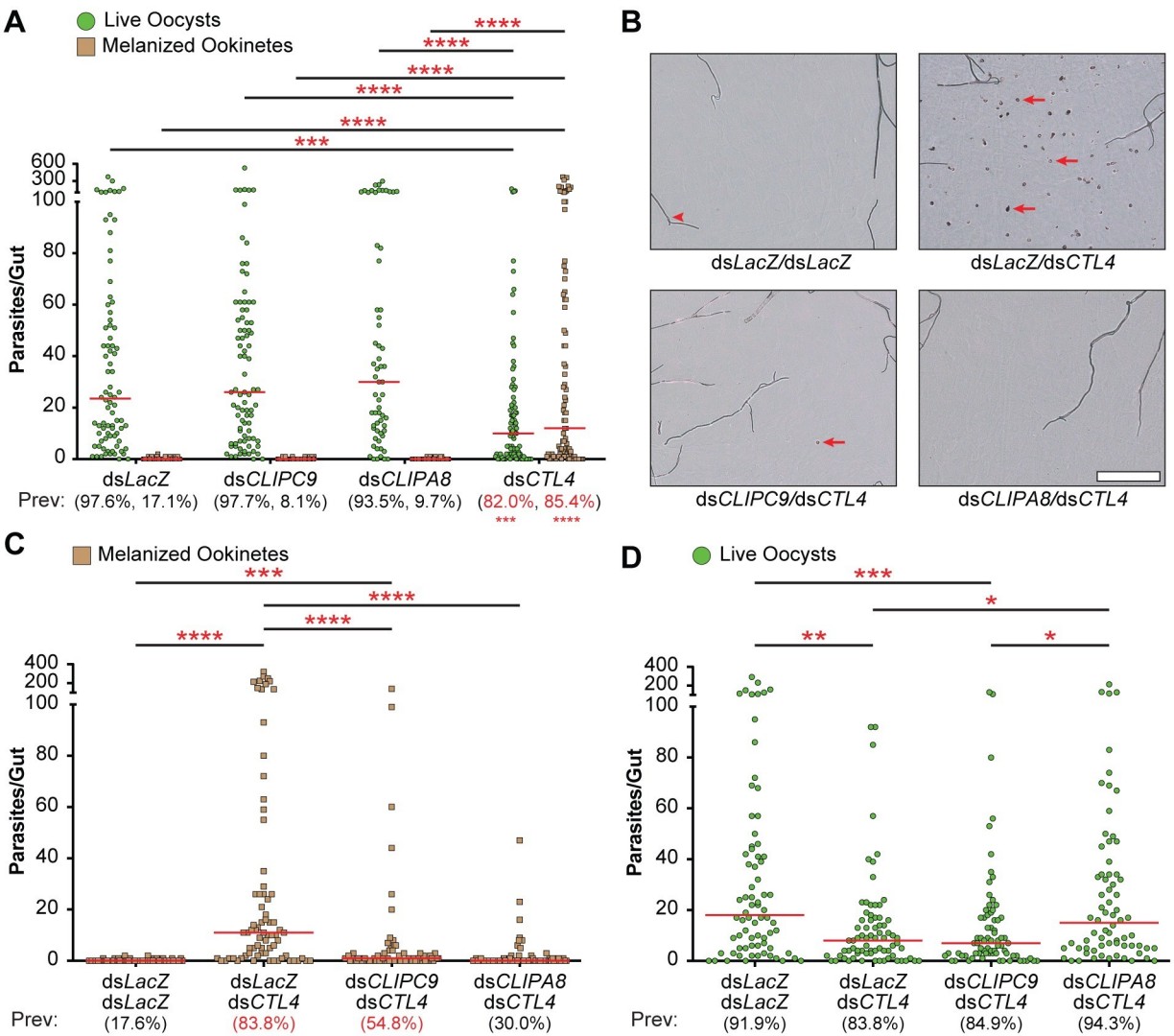

**Fig 5. CLIPC9 is required for *P. berghei* ookinete melanization.** (A) Single knockdown dot plots show the number of live oocysts (green dots) and melanized ookinetes (brown squares) present in each midgut. Red bars represent median values, which were compared using a Kruskal-Wallis test followed by Dunn's multiple comparisons test. Prevalence values (Prev) for GFP+ viable oocysts and melanized ookinetes are provided below each treatment group and were compared to ds*LacZ* control with Fisher's exact test. Significant differences are denoted in red. (B) Representative brightfield photomicrographs of mosquito midguts showing melanized ookinetes (red arrows) from the double knockdown analyses. Red arrowhead denotes a trachea. Scale bar is 100 μm. Co-silencing CLIPC9 and CTL4 reduces ookinete melanization (C) but does not rescue viable oocysts (D). Red bars represent median parasite values and were analyzed using a Kruskal-Wallis test followed by Dunn's multiple comparisons test. Prevalence values were analyzed with the Fisher's exact test and those significantly different than ds*LacZ*/ds*LacZ* control are in red. Asterisks denote statistical significance (*p ≤ 0.05, **p ≤ 0.01, ***p ≤ 0.001, ****p ≤ 0.0001). All data are pooled from three independent biological replicates.

around the dsRNA injection site near the thoracic anepisternal cleft (S4B Fig). This melanization response was specific to the CTL4 knockdowns, occurred in the absence of bacterial injection, and was absent from the contralateral, uninjected side of the thorax, suggestive of a wound-induced response. The ds*CTL4*-dependent thoracic melanization phenotype required both CLIPC9 and CLIPA8, as ds*CLIPC9*/ds*CTL4*- and ds*CLIPA8*/ds*CTL4*-treated groups had total spot areas comparable to control (S4C Fig) and 6- and 11-fold fewer mosquitoes with visible thoracic melanization relative to CTL4 knockdown alone, respectively (S4D Fig).

Collectively, these observations reveal that MelASA may be able to detect dsRNA-induced increases in melanization and suggest a novel role for CTL4/CTLMA2 in tempering excessive, wound-induced tissue melanization. However, we cannot rule out the possibility of an enhanced melanotic response in CTL4 knockdowns arising from a local infection seeded from the cuticle during dsRNA injection, as we recently reported that CTL4/CTLMA2 negatively regulates PPO activation following *E. coli* challenge [64]. Repeating these experiments with antibiotic-treated mosquitoes may help clarify if the ds*CTL4*-dependent thoracic melanization phenotype is associated with the melanization of microbes or wounded tissue. Regardless, these findings highlight another context in which CLIPC9 plays a critical role in the process of melanization.

### CLIPC9 is cleaved following *E. coli* challenge

CLIP-SPs are activated by a limited proteolytic cleavage targeting a site between the N-terminal CLIP domain and the C-terminal protease domain, and active CLIP-SPs circulate as 2-chain forms linked by an interchain disulfide bond [65]. To determine if CLIPC9 is present within the hemolymph compartment, we generated polyclonal antisera against full-length CLIPC9. We first performed non-reducing (Fig 6A, left) and reducing (Fig 6A, right) western analyses on hemolymph extracted from naïve mosquitoes and detected single proteins migrating at approximately 34 and 37 kDa, respectively. These bands closely correspond to the 35.7 kDa predicted MW for mature CLIPC9. Next, to determine if CLIPC9 is cleaved in response to wounding or septic infection, we performed reducing western analyses on hemolymph extracted from naïve, PBS-, and *E. coli*-challenged mosquitoes (Fig 6B). PBS challenge caused an early increase in circulating full-length CLIPC9, as indicated by the stronger band intensity at 60 min relative to the naïve control (Fig 6B). Following *E. coli* challenge, 2 CLIPC9 cleavage products migrating at approximately 30 and 12 kDa (hereafter referred to as p30 and p12, respectively) were found in the hemolymph in addition to the full-length protein (Fig 6B). The

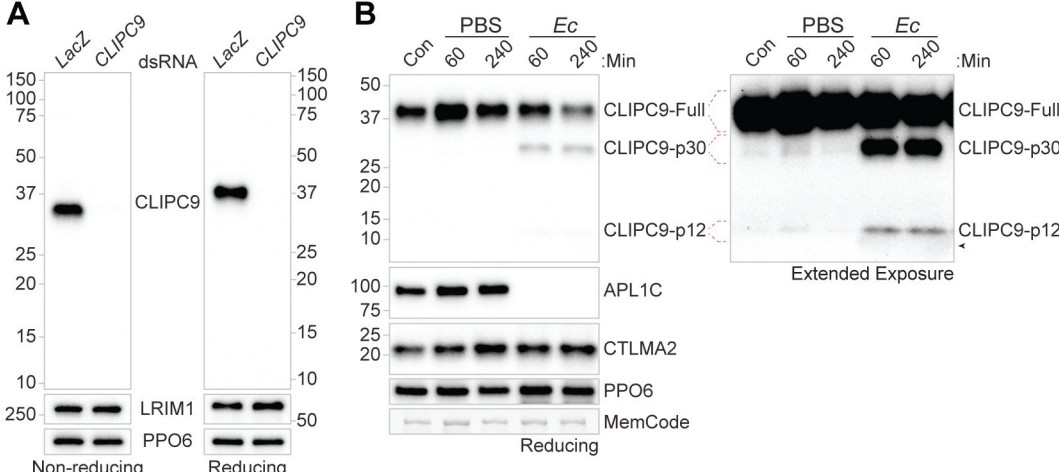

**Fig 6. CLIPC9 is a hemolymph protein cleaved after *E. coli* challenge.** (A) Western analysis of CLIPC9 in naïve hemolymph from ds*LacZ*- and ds*CLIPC9*-treated mosquitoes run on non-reducing (left) and reducing (right) SDS-PAGE. Blots were re-probed for LRIM1 and PPO6 to confirm equal loading. (B) Reducing western analysis of naïve/control hemolymph (Con) and hemolymph 60 and 240 min after PBS or *E. coli* (*Ec*) challenge. The anti-CLIPC9 extended exposure is provided to the right of the compilation to show the CLIPC9 p12 fragment. Blots were probed using antibodies targeting APL1C, CTLMA2, and PPO6 to confirm *E. coli* exposure, validate sample reduction, and confirm equal loading, respectively. Black arrowhead denotes a non-specific haze associated with *E. coli* injection occasionally observed below the p12 fragment in western analysis. Membranes were MemCode stained as an additional loading control. Images are representative of three independent biological replicates.

p12 fragment was consistently apparent following extended blot exposures and is shown to the right of the compilation. These fragment sizes are comparable to the 27.3 and 8.4 kDa products expected to be generated following CLIPC9 cleavage at its predicted cut site [51]. The p30 and p12 fragments were apparent at both 60 and 240 min post-challenge, with the latter timepoint characterized by the depletion of full-length CLIPC9. CLIPC9 cleavage following *E. coli* challenge coincided with reduction/loss of the *Anopheles Plasmodium*-responsive leucine-rich repeat 1-C (APL1C) protein western signal, which occurs due to limited proteolysis, as previously reported [52]. Only full-length CLIPC9 is observed under non-reducing western analysis of these samples, confirming the disulfide linkage joining the p30 and p12 fragments (S5 Fig). Moreover, the depletion of full-length CLIPC9 240 min post-*E. coli* challenge was not evident under non-reducing conditions suggesting that the disulfide-linked p30 and p12 fragments comprise the full-length protein. While amino acid sequencing of the p30 and p12 fragments will be necessary to fully characterize CLIPC9 limited proteolysis, our data support a cleavage event at or near the cut site predicted by Cao *et al.* [51]. We note faint cleavage products in naïve and PBS injected samples on extended blot exposure, suggesting low levels of basal and injury-associated CLIPC9 cleavage (Fig 6B). Collectively, these results indicate that CLIPC9 is potently cleaved following *E. coli* septic infection, and that the apparent depletion of full-length CLIPC9 is due to its conversion to a 2-chain form. The observation that full-length CLIPC9 depletion at 240 min was not coincident with increased p30 and p12 band intensities relative to the 60 min timepoint suggests the cleaved form may be degraded or localize to *E. coli* surfaces.

## CLIPC9 is regulated by mosquito complement

TEP1 functions upstream in the *An. gambiae* melanization cascade [4, 45–50], and its silencing strongly inhibits SPCLIP1 localization to microbial surfaces [46], and the cleavages of both CLIPA8 [46] and CLIPA28 [53]. Mature $TEP1_{cut}$ is stabilized through an interaction with the leucine-rich repeat containing proteins Leucine-Rich Immune Molecule 1 (LRIM1) and APL1C, which form a disulfide-linked heterodimer [66, 67]. Loss of the LRIM1/APL1C complex, driven by silencing either component, destabilizes $TEP1_{cut}$ and allows it to accumulate on self-tissue [66, 67]. $TEP1_{cut}$ accumulation recruits downstream complement components such as SPCLIP1 and CLIPA2 to self-tissue, leading to their depletion from the hemolymph in a TEP1-dependent manner [46, 48]. The comparable MelASA phenotypes observed in CLIPC9 and CLIP-SPH pathway knockdowns, led us to hypothesize that, like SPCLIP1, both CLIPC9 and CLIPA28 may similarly follow destabilized $TEP1_{cut}$ to self-tissues. In support, analysis of naïve hemolymph from LRIM1 knockdowns revealed the depletion of CLIPC9 and CLIPA28 along with SPCLIP1 and $TEP1_{cut}$ compared to the control (Fig 7A). We also note a subtle, but reproducible reduction in TEP1-Full following LRIM1 knockdown suggesting a low level of convertase activity. CLIPC9 and CLIPA28 reductions in the hemolymph were TEP1-dependent since co-silencing LRIM1 and TEP1 restored levels to baseline (Fig 7B). These hemolymph protein dynamics were not mirrored transcriptionally (S6 Fig) suggesting that CLIPC9 and CLIPA28 loss following LRIM1 knockdown are likely due to TEP1-dependent formation of a multi-protein complex on self-tissue. These results confirm the position of CLIPA28 downstream of mosquito complement and suggest that CLIPC9 is similarly regulated.

## SPCLIP1, CLIPA8, and CLIPA28 function in CLIPC9 cleavage

Recent work suggests that components of the positive regulatory CLIP-SPH cascade function upstream of several CLIPB PAPs [53]. The penultimate cascade positions for members of the CLIPC subfamily in other insects [9] and our finding that CLIPC9 is regulated by TEP1 led us

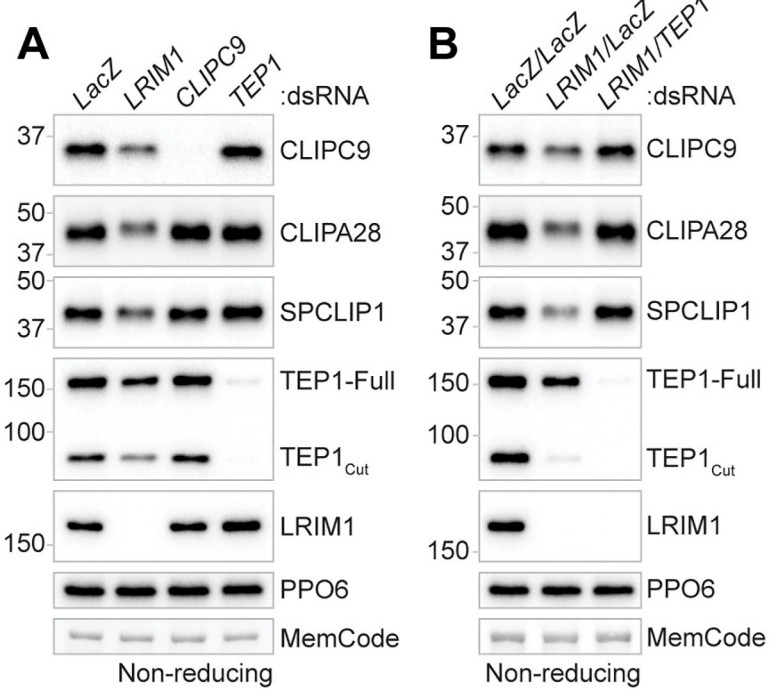

**Fig 7. CLIPC9 is regulated by TEP1.** Non-reducing western analysis of CLIPC9 and CLIPA28 in the hemolymph 3–4 d after (**A**) LRIM1 silencing or (**B**) LRIM1/TEP1 co-silencing. The depletion of TEP1cut and SPCLIP1 were tracked as positive controls for the effect of LRIM1 knockdown on mosquito complement components. Membranes were probed with a PPO6 antibody and stained with MemCode to confirm equal protein loading. Each image is representative of three independent biological replicates.

to explore the relationship of CLIPC9 with SPCLIP1, CLIPA8, and CLIPA28. For this, we utilized our *E. coli* challenge model to monitor CLIPC9 cleavage in CLIP-SPH pathway knockdowns. Reducing western analysis on whole hemolymph harvested 240 min after *E. coli* challenge revealed that the cleavage and apparent depletion of full-length CLIPC9 was attenuated by ds*SPCLIP1*, ds*CLIPA8*, and ds*CLIPA28* treatment relative to controls (Fig 8A–8C). The reduction of the p12 band was potent, whereas the p30 fragment was not, indicating an incomplete block of CLIPC9 cleavage. However, the reduction in CLIPC9 cleavage is functionally important since microbial melanization is inhibited in these knockdown backgrounds [5, 46, 53, 54]. The strikingly similar effect of SPCLIP1, CLIPA8, and CLIPA28 knockdown suggests that each of these proteins regulates CLIPC9 cleavage and positions this essential protease downstream of these CLIP-SPHs. In support, CLIPA28 cleavage was unaffected in CLIPC9 knockdowns, but potently blocked following ds*SPCLIP1* and ds*CLIPA8* treatments (Fig 8A–8C), as previously reported [53]. Collectively, these findings broaden our understanding of how the positive regulatory CLIP-SPH cascade regulates the *An. gambiae* melanization immune response (Fig 8D).

## CLIPC9 cleavage is enhanced in SRPN2 knockdowns

SRPN2 negatively regulates the *An. gambiae* melanization response, as its silencing promotes the melanization and killing of *P. berghei* ookinetes and the spontaneous formation of melanotic pseudotumors within the hemocoel [40]. We utilized the pseudotumor phenotype to confirm the ability of MelASA to detect negative regulatory proteins. The MelASA protocol was modified to align with pseudotumor formation by placing naive ds*LacZ*- and ds*SRPN2*-

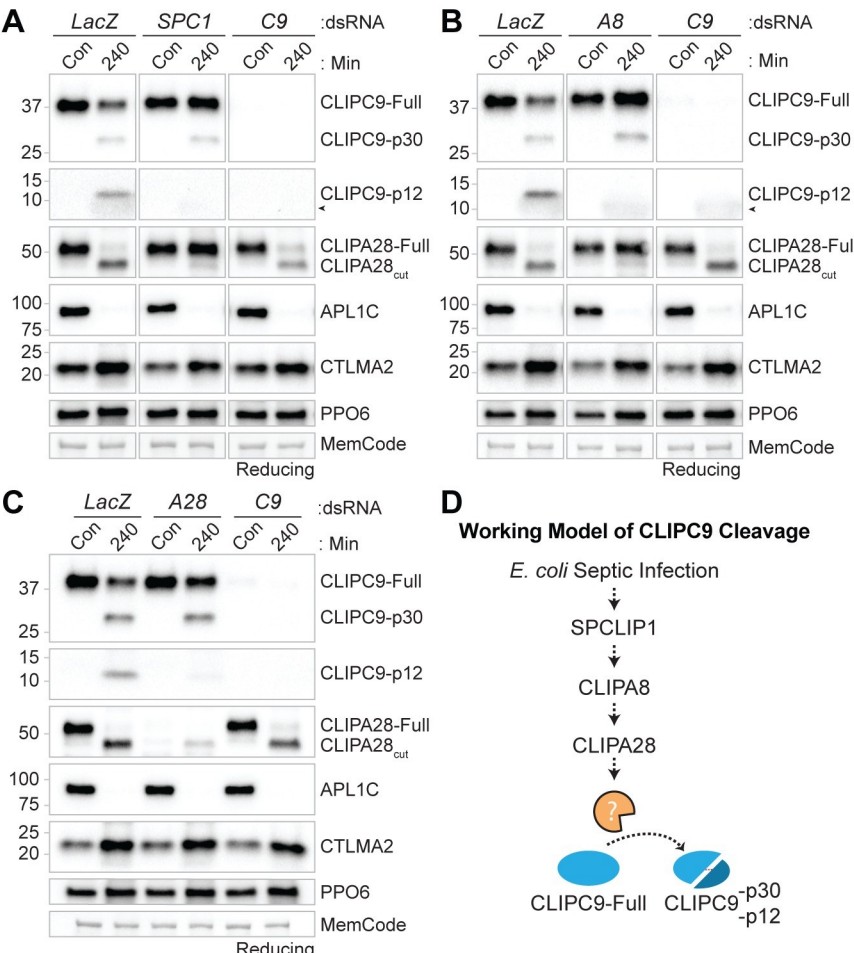

**Fig 8. SPCLIP1, CLIPA8 and CLIPA28 regulate CLIPC9 cleavage.** The impact of ds*SPCLIP1* (*SPC1*) **(A)**, ds*CLIPA8* (*A8*) **(B)**, and ds*CLIPA28* (*A28*) **(C)** treatment on CLIPC9 cleavage was assessed by reducing western analysis on naïve/control hemolymph (Con) and hemolymph collected 240 min after *E. coli* challenge. CLIPC9 knockdown samples are abbreviated (*C9*). The blots depicting the CLIPC9 p12 fragments in panels **(A-C)** were cropped from their respective anti-CLIPC9 extended exposures. The challenge, sample reduction, and equal protein loading were confirmed by probing against APL1C, CTLMA2, and PPO6, respectively. Membranes were MemCode stained as an additional loading control. Black arrowhead in panels **(A)** and **(B)** denotes a non-specific haze associated with *E. coli* injection occasionally observed below the CLIPC9 p12 fragment. Each western panel is representative of three independent experiments except for the CLIPA28 staining in panel **(A)** which was performed twice. **(D)** Cartoon illustration of CLIPC9 processing downstream of the CLIP-SPHs SPCLIP1, CLIPA8, and CLIPA28. Dashed arrows denote uncharacterized enzymatic steps.

treated mosquitoes into cups for 12 h following 4 d of gene knockdown, a timepoint when pseudotumors reach 100% prevalence [40]. Filter papers removed from SRPN2 knockdown cups consistently had numerous spots (Fig 9A top and S7 Fig), while the control filter papers lacked spots above threshold in 2 of 3 experimental replicates. Mosquito dissections performed following these assays confirmed the presence of melanotic pseudotumors in SRPN2 knockdowns (Fig 9A bottom). Collectively, these experiments support the established role of SRPN2 as a negative regulator of melanization, highlight the non-destructive nature of MelASA, and demonstrate that MelASA can be used to identify components involved in tissue melanization.

Given the enhanced cleavage of CLIPA8 [54] and CLIPA28 [53] following SRPN2 knockdown and our observation that CLIPC9 is cleaved downstream of these regulators, we asked if

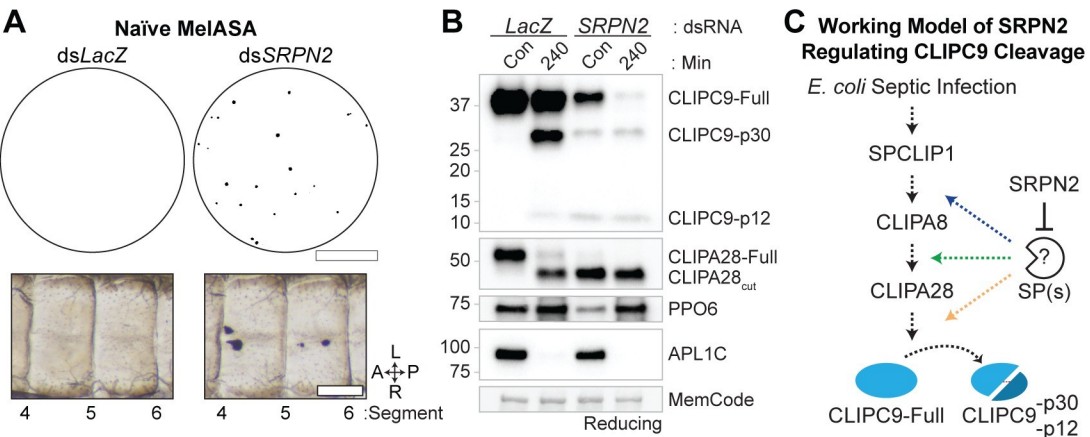

**Fig 9. SRPN2 silencing enhances CLIPC9 cleavage. (A)** Naïve Mini-MelASA performed 4 d after treatment with ds*LacZ* and ds*SRPN2* (Top). Filter papers were removed 12 h after mosquitoes were placed into cups. Images are representative of observations from three independent biological replicates. Scale bar is 1.3 cm. Dissections performed following Mini-MelASA confirmed the formation of melanotic pseudotumors in SRPN2 knockdowns (Bottom). Images span abdominal segments 4–6 with mosquitoes oriented along the anterior-posterior (A-P) and left-right (L-R) axes as indicated. Scale bar is 150 μm. **(B)** Western analysis of CLIPC9 cleavage in naïve SRPN2 knockdowns and ds*LacZ*-treated controls 4 d post-dsRNA administration (Con) or 240 min after *E. coli* challenge. The *E. coli* injection was confirmed by probing against APL1C and equal protein loading was confirmed by total protein MemCode staining. Image is representative of two independent biological replicates. **(C)** Working model of SRPN2 negatively regulating the serine protease(s) responsible for the cleavage-induced activation of CLIPC9 (orange arrow; this study), CLIPA28 (green arrow[53]; this study) and CLIPA8 (blue arrow[53, 54]). Dashed arrows represent uncharacterized enzymatic steps.

loss of SRPN2 would also drive CLIPC9 cleavage. We performed reducing western analysis on hemolymph from naïve and *E. coli* injected ds*SRPN2*- and ds*LacZ*-treated controls (Fig 9B). Treatment with ds*SRPN2* resulted in the generation of the CLIPC9 p30 and p12 cleavage products and a depletion of the full-length protein. The processing and depletion of CLIPC9 was concomitant with the near complete conversion of CLIPA28 to its cleaved form and the reduction of PPO6, as previously reported [53]. The CLIPC9 cleavage profile was notable in that depletion of the full-length protein was not coincident with increased p30 band intensity, suggesting that both full-length and cleaved CLIPC9 are depleted from the hemolymph due to either protein degradation or localization to melanotic pseudotumors. *E. coli* challenge potentiated this response and led to the further depletion of full-length CLIPC9 relative to its naïve control. Collectively, these results confirm a previous report that SRPN2 negatively regulates the CLIP-SPH cascade [53, 54] and support our previous results (Fig 8) positioning CLIPC9 downstream of this pathway (Fig 9C).

## Localization of CLIPC9 to *E. coli* requires CLIPA8

Microbial melanization is locally regulated to prevent off-target host damage through the specific recruitment of cascade components to microbial surfaces [44, 46, 48, 50]. To determine if CLIPC9 localizes to bacteria we utilized the *E. coli* bioparticle (*Ec*BP) challenge model (Fig 10A), as this system has been used to track the localization of melanization regulators [46, 48, 50] and produced virtually identical CLIPC9 cleavage results by whole hemolymph western as those obtained with live microbes (S8 Fig). We observed an enrichment of full-length CLIPC9 and cleaved p30 and p12 fragments on pellets obtained from *Ec*BP injected mosquitoes relative to pellets harvested from PBS injected controls (hereafter referred to as *Ec*BP pellets and PBS pellets, respectively; Fig 10B). The CLIPC9 p30 fragment observed in the *Ec*BP challenged soluble fraction could be derived from bioparticles that failed to pellet during sample processing

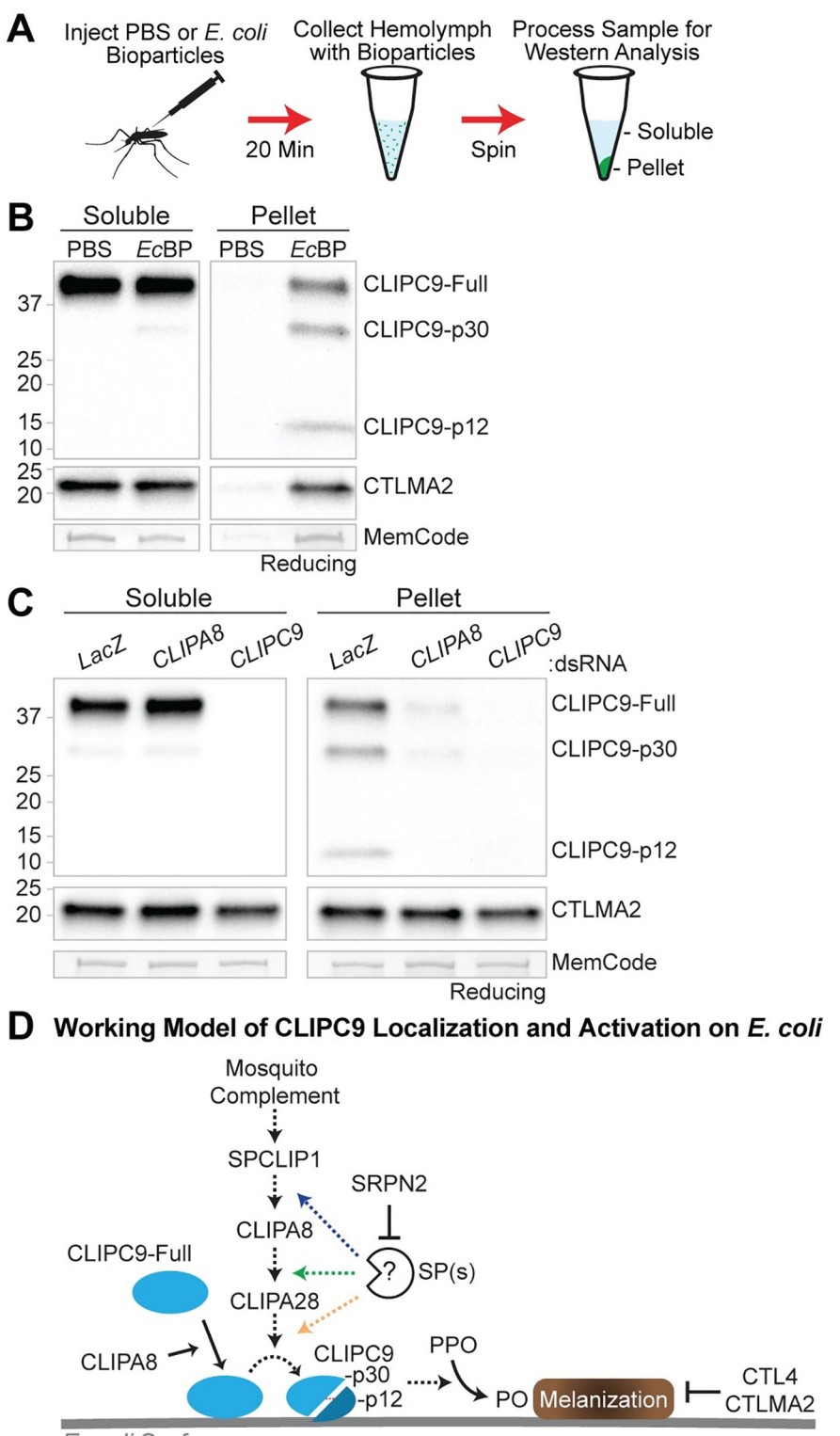

**Fig 10. CLIPA8 regulates the localization and cleavage of CLIPC9. (A)** The *E. coli* bioparticles (*Ec*BP) surface extraction model was utilized to assess the localization of CLIPC9 to bacterial surfaces 20 min post-challenge. PBS injection served as a negative control. **(B)** Reducing western analyses of CLIPC9 on soluble and pellet fractions following PBS or *Ec*BP injection. Blots were probed against CTLMA2 and MemCode stained to confirm sample reduction and protein loading, respectively. Western image is representative of three independent biological replicates.

(**C**) Reducing western analyses of CLIPC9 in soluble and pellet fractions obtained from ds*LacZ*-, ds*CLIPA8*-, and ds*CLIPC9*-treated mosquitoes following *Ec*BP injection. CTLMA2 staining was utilized as a localized immune protein loading control and to verify sample reduction. MemCode staining was used to confirm equal protein loading. Image is representative of experiments from three independent biological replicates. (**D**) Working model illustrating the CLIPA8-dependent recruitment of CLIPC9 to *E. coli* surfaces, its activation downstream of the mosquito complement-dependent CLIP-SPH pathway, and its negative regulation by SRPN2. The CTL4/CTLMA2 heterodimer negatively regulates *E. coli*-induced PO activity[64], however its relationships with mosquito complement, the CLIP-SPH pathway, and CLIPC9 are unknown. Dashed arrows indicate uncharacterized enzymatic steps.

or a minor pool of cleaved CLIPC9 that did not localize to the *Ec*BP pellet. The absence of the corresponding p12 band in the soluble fraction is likely due to it being below the limit of detection. In support, reducing western analysis on whole hemolymph collected 20 min post-*Ec*BP injection revealed the presence of the p30 fragment and absence of the p12 fragment (S9 Fig), while western analysis performed on concentrated *Ec*BP pellets harvested at the same time-point clearly show both cleavage products (Fig 10B). Collectively, these results suggest that CLIPC9 is recruited to microbial surfaces prior to its cleavage-induced activation. We also report that the C-type lectin CTLMA2 is enriched on *Ec*BP pellets, which corresponds well with its function in defense against gram-negative bacterial infection [61]. The specific enrichment of cleaved CLIPC9 on *Ec*BP pellets and its absence on PBS pellets support the *Ec*BP surface extraction model as a method for tracking the recruitment and cleavage of immune proteins. The faint bands observed from the PBS pellets likely indicate minor contamination from circulating hemocytes and/or fatbody liberated during hemolymph collection.

CLIPA8 may play a role in the recruitment of melanization components downstream of mosquito complement, as it is essential for PPO localization to *Beauveria bassiana* (*B. bassiana*) fungal hyphae [49]. Therefore, we hypothesized that the localization of full-length or cleaved CLIPC9 may require CLIPA8. Using the *Ec*BP model, we found that both full-length and cleaved CLIPC9 were strongly reduced on *Ec*BP pellets from ds*CLIPA8*-treated mosquitoes relative to ds*LacZ* controls (Fig 10C). CLIPA8 and CLIPC9 knockdown did not impact CTLMA2 localization, which confirmed equal loading and shows the specificity of the response. These results support a role for CLIPA8 in the recruitment of CLIPC9 to microbial surfaces where it can then be cleaved via limited proteolysis downstream of the mosquito complement-dependent CLIP-SPH pathway and ultimately contribute to PPO activation and microbial melanization (Fig 10D).

## Discussion

The arthropod melanization response requires the proteolytic conversion of PPO into active PO, which occurs downstream of sequentially activated CLIP-SPs and CLIP-SPHs. Biochemical studies in Lepidopteran and Coleopteran insect models indicate that CLIPAs regulate PPO activation by serving as co-factors for catalytic CLIPB PAPs [14, 22–32]. Recent work in *An. gambiae* suggests that some CLIPAs might function upstream in the melanization cascade by regulating CLIP-SP activation [53], however these relationships are unclear due to the lack of functional and biochemical data for members of the CLIPC subfamily, which activate CLIPBs in other insects [17–21]. Here, we report the discovery of CLIPC9 as a protease required for microbial melanization with the use of a novel screening method and describe its regulation downstream of the CLIP-SPHs SPCLIP1, CLIPA8, and CLIPA28.

Our finding that *An. gambiae* mosquitoes excrete a quantifiable amount of melanotic material following septic infection with *E. coli* led to the development of MelASA, which is an efficient method to functionally analyze the melanization response in this important disease vector. While the anatomical origin of this excreta is a focus of our future work, its associations

with both bacterial and spontaneous tissue melanization within the hemocoel lead us to speculate a connection with the renal excretory system, which facilitates the removal of hemolymph-borne metabolic wastes [68]. Regardless, quantitating this response via MelASA allowed for the identification of proteins required for melanization as well as a negative regulator of cascade activation. Recent improvements in the gene models of several insects including *M. sexta* [69], *D. melanogaster* [70], and *An. gambiae* [51] have facilitated phylogenetic comparisons and functional predictions for many CLIP-SPs and CLIP-SPHs, and our identification of CLIPC9 as a protease essential for microbial melanization exemplifies how MelASA can complement these bioinformatic approaches. In this regard, Cao *et al.* [70] phylogenetically positioned CLIPC9 close to *M. sexta* HP6 and *D. melanogaster* Hayan, 2 CLIPCs with reported roles in melanization.

The phylogenetic relationship of CLIPC9 with HP6 and Hayan suggest several possible mechanisms for how it may control septic infection-induced PPO activation and the melanization of *E. coli* and *P. berghei* ookinetes. The first possibility is that CLIPC9, like HP6, cleaves a CLIPB PAP. For example, *in vitro* reconstitution experiments performed by An *et al.* [18] demonstrated that HP6 cleaves and activates proPAP1, which had previously been demonstrated to activate PPO in the presence of 2 CLIP-SPHs [29]. The strong reduction in microbial melanization following CLIPC9 silencing suggests that it may be involved in the activation of multiple CLIPBs, as several are implicated in the melanization of abiotic beads [59], *P. berghei* ookinetes [5, 58], and self-tissue [42, 71]. The development of antisera against candidate CLIPBs and the production of recombinant proteins will facilitate the reverse genetic and *in vitro* reconstitution experiments required to identify CLIPC9 substrates and are strategies currently underway in our laboratory. Another possibility is that CLIPC9 functions similar to Hayan by directly cleaving PPO. Nam *et al.* [72] reported that Hayan mutants failed to cleave hemolymph PPO1 after a sterile or septic wound and demonstrated that the recombinant, active Hayan catalytic domain could cleave PPO1 *in vitro*. Biochemical experiments with recombinant CLIPC9 and purified PPO, akin to those performed by An *et al.* with CLIPB9 [42], will be essential for testing this possibility. Finally, our *P. berghei* experiments revealed that CLIPC9, similar to Hayan, may function in a melanization-associated darkening reaction, rather than microbial killing. An elegant genetic analysis by Dudzic *et al.* [73] indicated that Hayan is required for the darkening of wound sites in a reaction requiring PPO1 and PPO2, while another melanization pathway branch was required for PPO1 mediated defense against *S. aureus*. This complements an earlier study that revealed degrees of functional specialization among the 3 *D. melanogaster* PPOs [74]. We found that CLIPC9/CTL4 double knockdown reduced *P. berghei* ookinete melanization but did not rescue viable oocysts. In contrast, CLIPA8/CTL4 co-silencing potently inhibited ookinete melanization and rescued viable oocysts, supporting a previous report by Volz *et al.* [5]. These findings raise the possibility that specific PPOs downstream of CLIPA8 control CTL4 knockdown-induced ookinete killing, while other PPOs downstream of both CLIPA8 and CLIPC9 form a melanization pathway branch required for the darkening of killed parasites. While our biochemical data positioning CLIPC9 cleavage downstream of CLIPA8 support this possibility, our limited understanding of PPO activation during *An. gambiae* melanization reactions complicates this interpretation. While the slightly less potent block in ookinete melanization following CLIPC9/CTL4 co-silencing could be also be responsible for the failure to recover viable oocysts, we exercise caution with this interpretation, as the reported oocyst rescue data in co-silencing experiments with CTL4 vary widely and are not always commensurate with the reduction in ookinete melanization [5, 53]. Regardless, this finding is intriguing given the expanded, nine-membered PPO gene family in *An. gambiae* [75] and highlights the diverse roles these proteins may play in immune defense.

The identification and characterization of the CLIPs required for PPO activation may also provide new insights into the late-phase immune response targeting *Plasmodium* oocysts. Recent work from Kwon and Smith [76] using Keele strain *An. gambiae*, which are derived from the M-form genetic background, demonstrated that PPO2, PPO3, and PPO9 are required for the late-phase killing of *P. berghei* oocysts. The absence of melanization led the authors to speculate that the production of cytotoxic intermediates generated during the PO cascade drove oocyst killing. Interestingly, a study by Mitri *et al.* [77] using *An. coluzzii*, which are M-form and therefore genetically similar to the Keele strain, reported increased *P. falciparum* infection intensities following CLIPC9 knockdown. This study assessed mature oocyst burdens leaving open the prospect that CLIPC9 silencing compromised the late-phase response. The fact that late-phase immunity is implicated in defense against both *P. berghei* and *P. falciparum* [78] and our finding that CLIPC9 is required for septic infection-induced PPO activation raises the possibility that CLIPC9 is upstream of specific PPOs driving late-phase oocyst killing in M-form or M-form derived mosquitoes. This genetic association could explain the absence of a *P. berghei* oocyst phenotype in our CLIPC9 knockdown G3 strain mosquitoes. For example, Kwon *et al.* [79] reported that the transcription factors required for late-phase immunity differ between the Keele and G3 strains. It would be interesting to determine if CLIPC9 silencing in the Keele strain recapitulates the impaired late-phase response observed in PPO2, PPO3, and PPO9 knockdowns. Such a finding would suggest the possibility of strain-specific or infection stage-specific mechanisms governing CLIPC9 activation given that the late-phase response does not require TEP1 [79] and our observation that CLIPC9 is cleaved downstream of the TEP1-dependent CLIP-SPH pathway. Alternatively, a failure of CLIPC9 silencing to compromise late-phase immunity in the Keele strain could suggest a role for CLIPC9 in the activation of specific PPOs responsible for parasite darkening, but not oocyst killing given the absence of visibly melanized oocysts following the late-phase response [76]. An improved understanding of the protease cascades controlling *An. gambiae* melanization is likely to reveal exciting intersections with the PPOs driving late-phase immunity.

Our biochemical experiments positioning SPCLIP1, CLIPA8, and CLIPA28 upstream of CLIPC9 proteolysis is, to our knowledge, the first example of CLIP-SPHs modulating the cleavage of a CLIP-SP required for PPO activation. This report deepens our understanding of CLIPA function and supports recent speculation by El Moussawi *et al.* [53] that CLIPA28 and other CLIP-SPHs may control CLIP-SP activation. We found that SPCLIP1, CLIPA8, and CLIPA28 each play a striking similar role regulating CLIPC9 processing, supporting their participation in a common pathway. This interpretation is supported by the enhanced cleavage of CLIPA8 [53, 54], CLIPA28 [53], and CLIPC9 following SRPN2 silencing and we predict that that these CLIP-SPHs are required for the cleavage-induced activation of CLIPC9 by an upstream modular serine protease (Mod-SP). While there is no direct evidence in the literature that Mod-SPs require auxiliary factors to cleave CLIPCs, Gorman *et al.* [20] suggested that the *M. sexta* Mod-SP, HP14, may require a protein cofactor to cleave HP21 after *in vitro* reconstitution experiments with purified proteases did not reproduce the findings of Wang and Jiang [21], who demonstrated HP21 cleavage by HP14 using less pure samples. It would be interesting to determine if one or more regulatory CLIPAs were amongst the impurities. A recent study by Wang *et al.* [80] demonstrated that *An. gambiae* SP217 (AgSP217), which shares a similar domain arrangement with the melanization cascade initiating Mod-SPs in other insects, potentiated PO activity following addition to *M. sexta* hemolymph. Further investigation into the role of AgSP217 in *An. gambiae* melanization and assessing potential interactions with CLIPC9 and the CLIP-SPH cascade is warranted. It is curious that the CLIPC9 processing that did occur in SPCLIP1, CLIPA8, and CLIPA28 knockdowns was insufficient to activate PPO. The persistent cleavage indicates that some CLIPC9 proteolysis may be independent of

the CLIP-SPH pathway, that additional CLIPC9 regulators may await discovery, and reveals that CLIPC9 function post-cleavage may require at least one CLIPA co-factor. Given the extensive biochemical evidence that CLIPAs promote productive interactions between CLIPB PAPs and PPOs, it is possible that SPCLIP1, CLIPA8, and CLIPA28 facilitate the interactions required for CLIPC9 cleavage and its subsequent engagement with a downstream substrate. The inclusion of these CLIP-SPHs in the aforementioned *in vitro* reconstitution experiments will be vital for determining their respective contributions toward CLIPC9 activation and function.

The requirement of CLIPA8 for the recruitment of full-length and cleaved CLIPC9 to *E. coli* suggests an integral role for this CLIP-SPH in the localization and activation of downstream melanization cascade components. This finding also sheds light on how CLIPC9-dependent melanization is constrained to a microbial surface. It is unknown if full-length or cleaved CLIPA8 localizes to microbes, however Schnitger *et al.* [54] speculated that the faint appearance of cleaved CLIPA8 relative to the full-length protein on immunoblot may be due to the activated form localizing with PO and additional proteins to melanin coated pathogens. We do not know how CLIPA8 facilitates CLIPC9 recruitment to *E. coli* or if CLIPA8 activation precedes this event, however we speculate that it may involve physical interactions between these 2 proteins. A foundational study by Piao *et al.* [31] on the *Holotrichia diomphalia* CLIP-SPH, PPAF-II, revealed that its CLIP domain contained a central cleft essential for physical interactions with cleaved PO, while the protease-like domain contained 2 clefts speculated to serve as docking sites for additional proteins. CLIPA8 is required for PPO recruitment to *B. bassiana* hyphae [49] raising the intriguing possibility that it may also facilitate the localization of downstream proteases such as CLIPC9. Immunolocalization experiments assessing if CLIPA8 associates with microbial surfaces and interaction studies with full-length and activatable wildtype and site-directed mutant proteins will be essential for clarifying these putative relationships.

CLIPA8 and CLIPA28 are cleaved downstream of the mosquito complement components TEP1 and SPCLIP1 [46, 53], which are both reported to localize to microbial surfaces [44, 46]. The requirement of TEP1 and SPCLIP1 for defense against bacteria and *Plasmodium* [4, 81, 82] and the lack of reported phenotypes in CLIPA8 and CLIPA28 knockdowns [5, 53, 54], suggests that mosquito complement is involved in multiple effector responses, while CLIPA8 and downstream factors play specialized roles in melanization. This is supported by our finding that CLIPC9 silencing did not increase susceptibility to *E. coli* or enhance *P. berghei* infection intensity. Work in *D. melanogaster* indicates that the importance of melanization in immune defense is microbe-specific [83]. Indeed, the requirement of CLIPA8 and CLIPA28 for defense against *B. bassiana* suggests a key role for melanization in antifungal defense [49, 53]. We speculate that CLIPC9 may also be required for defense against *B. bassiana*, and we are actively exploring this possibility. We propose a model where CLIPA8 activation downstream of TEP1 and SPCLIP1 demarcates a point of cascade specialization where active CLIPA8 regulates the melanization immune response by promoting CLIPC9 localization and subsequent activation downstream of the CLIP-SPHs SPCLIP1, CLIPA8, and CLIPA28.

This report describes CLIPC9 as the first CLIPC involved in the mosquito melanization immune response and demonstrates its essential role in septic infection-induced PO activity and the melanization of *E. coli* and *P. berghei* ookinetes. Our biochemical experiments positioning CLIPC9 proteolysis downstream of SPCLIP1, CLIPA8, and CLIPA28 indicate that these CLIP-SPHs may regulate the activation of catalytic CLIP-SPs. Our data additionally suggest that CLIPA8 plays a role in the recruitment of proteins specifically required for melanization to microbial surfaces. This study demonstrates the utility of MelASA as a streamlined approach to functionally study melanization, which should prove useful in the quest to understand and ultimately tailor mosquito immunity.

## Materials and methods

### Ethics statement

All animal studies were performed under Institutional Animal Care and Use Committee approved protocols and in accordance with the guidelines of the Institutional Animal Care and Use Committee of the University of Pennsylvania (IACUC, protocol 806192). Prior to mosquito blood feeding, *P. berghei* infected mice were anesthetized with a mixture of ketamine and xylazine. Following the experiment, mice were humanely euthanized with $CO_2$ or cervical dislocation. Euthanasia was confirmed in accordance with our IACUC approved guidelines.

### *Anopheles gambiae* rearing

The following reagent was obtained through BEI Resources, NIAID, NIH: *Anopheles gambiae*, Strain G3, Eggs, MRA-112, contributed by Mark Q. Benedict. Mosquitoes were reared at 28˚C and 75% relative humidity on a 12 h light/dark cycle. Adults were maintained on 10% sucrose and provided heparinized sheep blood (HemoStat Laboratories) to stimulate egg production. Eggs were surface sterilized with 1% bleach prior to floatation in water supplemented with a 20 mg/mL baker's yeast solution. Prior to pupation, larvae were fed ground fish food (Tetramin) and dry cat chow (Friskies) and maintained at a density of 300 larvae per L.

### Double-stranded RNA production and gene silencing

All kits and reagents were used following instructions provided by the manufacturer. Double-stranded RNA (dsRNA) was generated from T7 RNA polymerase promoter-tagged templates. T7-tagged templates were generated with the iProof High-Fidelity PCR Kit (Bio-Rad) using sequences cloned into plasmids pIB (LacZ), pQE30 (*CLIPC9* and *CLIPA28*), and pIEx10 (*CLIPA8*, *SPCLIP1*, and *LRIM1*), or mosquito cDNA (*CTL4*, *TEP1*, *SRPN2*, *CLIPC1*, *CLIPC2*, *CLIPC3*, *CLIPC4*, *CLIPC6*, *CLIPC7*, *CLIPC10*) as templates. Amplicons were purified with the GeneJET PCR Purification Kit (Thermo Fisher Scientific) and validated by size using gel electrophoresis. The HiScribe T7 High Yield RNA Synthesis Kit (NEB) was used to generate dsRNA from the purified T7-tagged amplicons. Reaction products were purified with the GeneJET RNA Purification Kit (Thermo Fisher Scientific), dried with a SpeedVac (Savant), and reconstituted to 3 μg/μL with ultrapure water. T7 primer sequences are provided in the Supporting Information (S1 Table).

Single gene knockdown experiments were performed by microinjecting $CO_2$ anesthetized mosquitoes with 69 nL of 3 μg/μL dsRNA (207 ng/mosquito). Co-silencing experiments were performed by injecting 138 nL of a 1:1 mixture of 3 μg/μL dsRNAs (414 ng/mosquito) targeting each transcript. For example, the control in co-silencing experiments was injected with 138 nL of ds*LacZ*, while single and double knockdowns were injected with 138 nL of 1:1 mixtures of ds*LacZ*/target dsRNA and target dsRNA/target dsRNA, respectively. Experiments were performed 3–4 d after dsRNA injections. Knockdowns were validated by qRT-PCR. Briefly, whole mosquito RNA was isolated from 10 mosquitoes per treatment group using TRIzol (Thermo Fisher Scientific). Purified total RNA was treated with TURBO DNase (Thermo Fisher Scientific), and cDNA was generated from 1 μg total RNA using the iScript cDNA Synthesis Kit (Bio-Rad). Assays were performed with PerfeCTa SYBR Green SuperMix, Low ROX (Quanta BioSciences) on a QuantStudio 6 Flex Real-Time PCR System (Applied Biosystems). Gene expression was analyzed by the comparative Ct method with values normalized to the S7 ribosomal protein gene. Average gene silencing efficiencies (± SD) are provided in the Supporting Information (S2 and S6 Figs) except for CTL4 (n = 6) and SRPN2 (n = 3), which were $0.11 \pm 0.05$ and $0.33 \pm 0.18$, respectively. Primer sequences for qRT-PCR analysis are provided in the Supporting Information (S2 Table).

### Bacterial strains and preparation for mosquito injection

Live *E. coli* infections were performed with a DH10β strain expressing green fluorescent protein (GFP) and carrying ampicillin resistance. Prior to 69 nL microinjection, bacteria were grown to mid-log phase, rinsed 3 times in 1x PBS, and resuspended to OD 0.8. Cultures were concentrated to the theoretical ODs 3.2, 6.4, or 32 by pelleting the OD 0.8 suspensions and removing appropriate volumes of the cell-free supernatants. The OD 3.2, 6.4, and 32 values corresponded to average doses of $7.7 \times 10^5$, $1.3 \times 10^6$ and $7.9 \times 10^6$ CFUs/μL, respectively.

### Melanization-associated Spot Assay

MelASA was performed in Standard (50 mosquitoes/group) and Mini (20 mosquitoes/group) formats. All experimental parameters, apart from the cup dimensions, filter paper diameters, and group sizes, were identical between formats. For challenge-associated MelASAs, wild-type or dsRNA treated mosquitoes were injected with PBS or *E. coli* OD 6.4 and recovered for 4 or 12 h in netted cups containing freshly inserted P8 grade, white filter papers (Fisherbrand). SRPN2 knockdown MelASAs were performed 4 d after dsRNA administration without challenge. Naïve CTL4 knockdown MelASAs were performed on filter papers left in the housing cups during the 4 d dsRNA incubation. Standard MelASAs were performed at 4 and 12 h with 16 oz, 3 inch bottom diameter soup cups with inserted 9 cm diameter filter papers. Mini-Mel-ASAs were conducted at 12 h with 3 oz, 1.5 inch bottom diameter cups bottom-lined with 6 cm diameter filter papers. At the indicated timepoints, filter papers were removed and imaged in a Bio-Rad ChemiDoc MP System under white epi-illumination without filters using Image Lab version 5.2.1 software (Bio-Rad). Image processing and spot area quantification was performed in Fiji is Just Image J (FIJI, Version 2.00-rc-68/1.52h) [84]. A schematic outlining this procedure and the accompanying FIJI macros are provided in the Supporting Information (S1A Fig). Briefly, ChemiDoc files were exported as TIFs and uniformly processed in bulk with macro #1 and then macro #2 to generate thresholded images. The thresholded images were converted to masks, and the total area of melanotic excreta on the filter papers was then measured using the Analyze Particles function. The total filter paper spot area for the control group was set to 1.0 and the experimental groups were adjusted accordingly for statistical analysis.

### Bacterial melanization and survival assays

PO enzyme activity was measured 4 h after the injection of PBS or *E. coli* OD 6.4. Hemolymph was collected from cold anesthetized mosquitoes by the proboscis clipping method directly into 1X PBS containing 2X EDTA-free cOmplete Mini Protease Inhibitor Cocktail (Roche). Hemolymph protein concentrations were determined using a modified Bradford assay (Bio-Rad) and standardized to 6 μg total hemolymph protein per sample. The absorbance at 492 nm was measured as a proxy for PO enzyme activity every 10 min for a total of 90 min after the combination of hemolymph protein and 3 mg/mL L-DOPA. Terminal measurements at 90 min were compiled for statistical analysis.

The effect of *E. coli* septic infection on *An. gambiae* survival was tracked for 7 d after the injection of bacteria at OD 3.2. This dose was also utilized to assess the intensity of melanization along the ventral surface of the dorsal cuticle, which was blindly scored 2 d after bacterial injection. Cuticles were dissected, incubated in 4% paraformaldehyde at 4°C overnight, rinsed in 1X PBS, mounted in Fluoromount-G (SouthernBiotech) and rated as containing no, very mild, mild, moderate or severe melanization. Cuticle scores for each treatment group were then pooled for statistical analysis. Images were obtained using a Zeiss Stemi 305 stereo microscope with an Axiocam 105 color camera with the Zen 2 blue edition software. Cuticle

dissections to confirm melanotic pseudotumors in SRPN2 knockdowns were performed as described above.

### *Plasmodium berghei* infections

The interactions between *An. gambiae* and *Plasmodium* parasites were assayed using the rodent malaria model, *P. berghei*. The following reagent was obtained through BEI Resources, NIAID, NIH: *Plasmodium berghei*, Strain (ANKA) 507m6cl1, MRA-867, contributed by Chris J. Janse and Andrew P. Waters. Low passage *P. berghei* frozen stocks were serially passaged in 6–8 week-old Swiss Webster mice by intraperitoneal injection. Mouse blood parasitemia and microgametocyte exflagellation were assessed by microscopy prior to mosquito blood feeding. Mosquitoes were fed on anesthetized mice carrying a 2–6% parasitemia 3 d post-infection. Mosquitoes were fed at 21°C and held at this temperature for the duration of the experiment. Midguts were isolated from mosquitoes 9–10 d post-bloodmeal, incubated in 4% paraformaldehyde for 60–90 min at room temperature, rinsed in 1X PBS, and mounted in Vectashield (Vector Laboratories). GFP-positive oocysts and melanized ookinetes were manually counted and imaged with a Leica DM6000 widefield fluorescence microscope. Raw oocyst and ookinete counts are provided in the Supporting Information (S1 Dataset).

### Cloning and polyclonal antibody development

The full-length *CLIPC9* and *CLIPA28* nucleic acid sequences without the endogenous signal peptides BamHI and HindIII restriction cloned into pQE30 vectors with C-terminal 6X His tags. Removal of the *CLIPC9* signal peptide sequence was based on the updated annotation by Cao *et al.* [51]. Sequence validated vectors were inserted into *E. coli* M15 for IPTG-induced protein expression. Proteins induced by 1 mM IPTG entered inclusion bodies and were extracted with 8 M urea and purified with TALON Metal Affinity Resin (Takara) under denaturing conditions. Purified proteins were sent to Cocalico Biologicals (Stevens, PA 17578) for rabbit polyclonal antisera production. Polyclonal antiserum against CLIPC9 was affinity purified against full-length CLIPC9 generated from a baculovirus-induced expression system in *T. ni* cells. Briefly, *CLIPC9* was subcloned into a pFastbac1 vector with a C-terminal 6X His tag. Recombinant CLIPC9 baculovirus was expressed in *T. ni* cells grown in ESF-921 media (Expression Systems). Conditioned media was collected 60 h post-infection, concentrated, and diafiltrated with tangential flow filtration, and affinity purified using TALON Metal Affinity Resin. Column eluates were further purified by ion-exchange and size-exclusion with an AKTA Pure chromatography system. Purified CLIPC9 protein was immobilized using an AminoLink Immobilization Kit and the polyclonal antiserum was purified according to the manufacturer provided instructions (Thermo Scientific). Polyclonal antiserum against CLIPA28 was used crude. Cloning primers are provided in the Supporting Information (S3 Table).

### Western analyses

Unless otherwise indicated, hemolymph was harvested from cold anesthetized naïve, *E. coli*, or *E. coli* bioparticles injected mosquitoes by the proboscis clipping method directly into 2X non-reducing SDS PAGE sample buffer (Pierce). Experiments assessing the septic infection-induced cleavage of CLIPC9 used viable *E. coli* at OD 32 or killed *E. coli* pHrodo Red (Invitrogen) conjugated bioparticles at 20 mg/mL. Bioparticles were resuspended in sterile 1X PBS, vigorously vortexed, and sonicated before injection. Hemolymph was collected 20, 60, or 240 min after injection. When indicated, samples were reduced by the addition of Bond-Breaker TCEP Solution (Thermo Scientific) to a final concentration of 25 mM and heated to 95°C for

5 min. Samples were run on precast 4–15% or Any kD Mini-PROTEAN TGX gels (Bio-Rad). Proteins were transferred under semi-dry conditions using a Trans-Blot Turbo System (Bio-Rad) to PVDF membranes, total protein MemCode stained according to the manufacturer provided instructions, blocked in 3% milk for at least 1 h at room temperature, and probed with primary antisera for 1 h at room temperature or overnight at 4˚C. Rabbit anti-CLIPC9, -CLIPA28, -APL1C-CT tail, -LRIM1, -SPCLIP1, -TEP1, and -PPO6 were used at 1:500, 1:500, 1:2000, 1:1000, 1:1000, 1:1000, and 1:2000, respectively. Rat anti-CTLMA2 was used at 1:250. Rabbit and rat primary antibodies were detected with horseradish peroxidase (HRP) conjugated anti-rabbit (Promega) and anti-rat (Jackson ImmunoResearch) secondary antibodies, respectively. Membranes were developed with Clarity ECL Substrate (Bio-Rad) and imaged using the ChemiDoc MP System (Bio-Rad). Average exposure times ± SD for the whole hemolymph westerns tracking both CLIPC9-Full and CLIPC9-p30 were 16.1 ± 4.6 sec. CLIPC9-p12 detection required an extended exposure time of 532.0 ± 191.4 sec.

### Bacterial bioparticle surface extractions

The accumulation of hemolymph proteins onto bacterial surfaces was assessed as previously described [46] with minor modifications. Mosquitoes were injected with 69 nL *E. coli* pHrodo Red conjugated bioparticles at 20 mg/mL. Hemolymph collection buffer was prepared by dissolving one cOmplete Mini Protease Inhibitor Cocktail tablet in 7 mL of 15 mM pH 8.0 tris buffer. Hemolymph was collected into protein LoBind tubes (Eppendorf) from 60 cold anesthetized mosquitoes 20 min post-injection into 48 μL collection buffer. Bioparticles were then separated by centrifugation at 6000 x g for 4 min at 4˚C. The supernatant was transferred into a new LoBind tube and supplemented with 12 μL 5X non-reducing SDS PAGE sample buffer. The bioparticle-enriched pellet fraction was washed by resuspending the pellet in 400 μL collection buffer and transferred to a new LoBind tube. The suspension was again centrifuged at 6000 x g for 8 min at 4˚C. The final supernatant was discarded, and the pellet was resuspended in 20 μL 2X non-reducing SDS PAGE sample buffer. Samples were reduced and analyzed by western as indicated above.

### Statistical analyses

All statistical analyses were performed with GraphPad Prism Version 8.0.0. Data were assessed for normality using Shapiro-Wilk tests prior to statistical analyses. Normally distributed and non-normally distributed data were analyzed with parametric and non-parametric tests, respectively, as indicated in the figure legends.

### Supporting information

**S1 Fig. Filter paper image processing workflow.** **(A)** Filter paper image processing and analysis pipeline to generate and quantify thresholded images in FIJI[84]. **(B)** Unprocessed (top row, grayscale) and processed (bottom row, thresholded) filter papers from 50 mosquitoes 12 h after PBS or *E. coli* injection. Scale bar is 2.5 cm.
(TIF)

**S2 Fig. Experimental gene knockdown analyses.** **(A)** CLIPC screen gene expression analysis 3–4 d after dsRNA administration. **(B)** Cross-silencing analysis of the CLIPC candidates and the CLIP-SPH pathway members following ds*CLIPC9* treatment. CLIPC9 gene expression was included as a positive control. **(C)** Cross-silencing analysis of CLIPC9 following ds*CLIPA8*, ds*SPCLIP1*, and ds*CLIPA28* treatments. Expression values for CLIPA8, SPCLIP1, and CLIPA28 were included as positive controls. All mean qRT-PCR expression values are relative

to their respective ds*LacZ* treatment groups, which are set to 1.0 and denoted by the red dotted line. All error bars are ± SD. All data are pooled from three independent biological replicates. (TIF)

**S3 Fig. CLIPC9 silencing does not affect survival following *E. coli* septic infection.** Mosquito survival was tracked for 7 d following septic infection with *E. coli* at OD 3.2 in ds*LacZ*-, ds*CLIPC9*-, and ds*TEP1*-treated mosquitoes. TEP1 knockdowns were included as a positive control for reduced survival following *E. coli* infection[48]. Experimental groups were compared to the ds*LacZ* group with the Log-rank test. Asterisks denote statistical significance (****p ≤ 0.0001). Curves are averages from three independent biological replicates and dotted lines are 95% confidence intervals. (TIF)

**S4 Fig. ds*CTL4* causes an enhanced injection site melanization response that requires CLIPC9 and CLIPA8.** Naïve standard MelASAs performed on filter papers removed from **(A)** single or **(C)** double gene knockdown cups 3–4 d after dsRNA injection (one sample t-tests). Final group sizes differed slightly due to typical mosquito attrition, however intraexperimental groups never differed by more than 4 individuals. Error bars are ± SD. Asterisks denote statistical significance (*p ≤ 0.05). Brightfield photomicrographs of mosquito thoraxes 3–4 d after **(B)** single or **(D)** double gene knockdown. Single gene knockdown data were compared to ds*LacZ* control using Fisher's exact test. The ds*LacZ*/ds*LacZ* treatment group was excluded from the double gene knockdown analysis. The ds*CLIPC9*/ds*CTL4* and ds*CLIPA8*/ds*CTL4* treatment groups were each compared to the ds*LacZ*/ds*CTL4* group using Fisher's exact test. The regions of injection-induced melanization are outlined in white. Scale bars are 2.5 cm in **(A)** and **(C)**, 500 μm in **(B)**, and 200 μm in **(D).** All data are compiled from three independent biological replicates. Abbreviations: ns = not significant; nd = not determined. (TIF)

**S5 Fig. CLIPC9 cleavage products are disulfide linked.** The septic infection-induced cleavage of CLIPC9 was assessed by non-reducing western analysis of naïve/control hemolymph (Con) and hemolymph 60 and 240 min after PBS or *E. coli* (*Ec*) challenge. The anti-CLIPC9 extended exposure is provided to the right of the compilation to confirm the absence of the CLIPC9 p12 fragment. Blots were probed with antibodies against APL1C and PPO6 to confirm *E. coli* exposure and equal protein loading, respectively. Black arrowhead denotes a non-specific haze associated with *E. coli* injection in western analysis. Membranes were MemCode stained as an additional loading control. Image is representative of three independent biological replicates. (TIF)

**S6 Fig. Gene expression analysis in LRIM1 and TEP1 knockdowns.** Mean qRT-PCR expression analysis of CLIPC9, CLIPA28, and SPCLIP1 3–4 d after ds*LRIM1* and ds*TEP1* administration. Expression values for LRIM1 and TEP1 were included as positive controls. Expression values are relative to the ds*LacZ* treatment group, which is denoted by the red dotted line set to 1.0. Error bars are ± SD. Experimental replicates are from three independent mosquito generations. (TIF)

**S7 Fig. Enhanced MelASA spot generation in ds*SRPN2*-treated mosquitoes.** Additional naïve Mini-MelASA replicates 4 d after gene knockdown in ds*SRPN2*- and ds*LacZ*-treated mosquitoes. Filter papers were removed after 12 h. Scale bar is 1.3 cm. (TIF)

**S8 Fig. CLIPC9 is cleaved after *E. coli* bioparticle injection.** Challenge-induced cleavage of CLIPC9 was assessed by reducing western analysis on whole hemolymph 60 and 240 min after

PBS or *E. coli* bioparticle (*Ec*BP) injection and from naïve/control (Con) mosquitoes. The anti-CLIPC9 extended exposure is provided to the right of the compilation to show detection of the CLIPC9 p12 fragment. Blots were probed against APL1C, CTLMA2, and PPO6 to confirm the *Ec*BP challenge, adequate sample reduction, and equal protein loading, respectively. Membranes were MemCode stained as an additional loading control. Western image is representative of experiments performed on two independent biological replicates.
(TIF)

**S9 Fig. CLIPC9 p12 fragment is detectable 60 min after *E. coli* bioparticle injection.** Challenge-induced cleavage of CLIPC9 was assessed by reducing western analysis on whole hemolymph 20 and 60 min after *E. coli* bioparticle (*Ec*BP) injection and from naïve/control (Con) mosquitoes. The anti-CLIPC9 extended exposure is provided to the right of the compilation to show detection of the CLIPC9 p12 fragment. Blots were probed against PPO6 to confirm equal protein loading. Western image is representative of four independent biological replicates.
(TIF)

**S1 Table. T7 primers used for dsRNA template synthesis.**
(DOCX)

**S2 Table. Primers used for qRT-PCR analysis.**
(DOCX)

**S3 Table. Cloning primers used for protein production.**
(DOCX)

**S1 Dataset. Counts of live oocysts and melanized ookinetes in single and double gene knockdown *P. berghei* experiments.**
(XLSX)

## Acknowledgments

We thank Dr. Igor E. Brodsky, Dr. Sara Cherry, Dr. Christopher A. Hunter, Dr. Michael Atchison, Dr. Michael J. May, and Dr. Bruce D. Freedman for helpful scientific discussions and for mentorship to GLS; Abigail R. McCrea for maintaining the G3 mosquito colony; the University of Pennsylvania School of Veterinary Medicine ULAR staff for assistance with mouse husbandry; Dr. Gordon Ruthel of the Penn Vet Imaging Core for providing valued advice on mosquito midgut imaging; Dr. Leslie B. King, Dr. Sutopa B. Dwivedi, and Sarah D. Sneed for scientific discussion and editorial comments.

## Author Contributions

**Conceptualization:** Gregory L. Sousa.

**Data curation:** Gregory L. Sousa.

**Formal analysis:** Gregory L. Sousa, Michael Povelones.

**Funding acquisition:** Richard H. G. Baxter, Michael Povelones.

**Investigation:** Gregory L. Sousa.

**Methodology:** Gregory L. Sousa.

**Project administration:** Michael Povelones.

**Resources:** Ritika Bishnoi, Richard H. G. Baxter.

**Supervision:** Richard H. G. Baxter, Michael Povelones.

**Visualization:** Gregory L. Sousa, Michael Povelones.

**Writing – original draft:** Gregory L. Sousa, Michael Povelones.

**Writing – review & editing:** Gregory L. Sousa, Ritika Bishnoi, Richard H. G. Baxter, Michael Povelones.

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
