## [Decision Letter · Decision Letter 0]

21 Aug 2020

Dear Dr. Povelones,

Thank you very much for submitting your manuscript "The CLIP-domain serine protease CLIPC9 regulates melanization downstream of SPCLIP1, CLIPA8, and CLIPA28 in the malaria vector Anopheles gambiae" for consideration at PLOS Pathogens. As with all papers reviewed by the journal, your manuscript was reviewed by members of the editorial board and by several independent reviewers. The reviewers appreciated the attention to an important topic. Based on the reviews, we are likely to accept this manuscript for publication, providing that you modify the manuscript according to the review recommendations.

Your manuscript was evaluated by three reviewers and me. The clear consensus is that this manuscript is a significant scientific contribution because of (i) its mechanistic discoveries on the melanization pathway of a mosquito that is a major vector of malaria in Africa, and (ii) its presentation of a novel, non-invasive method for measuring phenoloxidase activity in mosquitoes.

When revising your manuscript, please pay attention to the reviewer comments.

Reviewer 1 requests elaboration on the CLIPC9 RNAi survival phenotype. The manuscript already draws parallels to what has been published for CLIPA8, but elaboration on this point is warranted. Reviewer 1 also recommends an experiment that you should keep in mind when conducting follow-up studies (so, not required for this submission).

Reviewer 2 presents some technical questions on spontaneous melanization and several Western blots, plus the strength of the model. I believe all of these issues can be addressed in the revised manuscript and rebuttal letter (more on this below).

Reviewer 3 asks for further explanation or clarification on some aspects of the interpretation of the data. Note that reviewers 2 and 3 shared some concern about the interpretation of CLIPC9 cleavage, based on the p12/p30 Western blots. Clarifying this point will be essential in your revision.

In addition, I have several minor comments. On more than one occasion the manuscripts extrapolates the E. coli findings to all Gram(-) bacteria (for example, lines 181-182, 193). This claim is not warranted and should be deleted. To make this claim, the study would have to examine several Gram(-) bacteria and obtain similar findings for all of them. I also recommend that you not use the term “master regulator” (as in the submission letter) or “master negative regulator” (line 330-331) because master regulators are classically defined as transcription factors related to cell differentiation. Finally, multiple statistical tests were used, and they are clearly detailed in the figure legends. I imagine that they were selected after testing the data for normality. If that is the case, include a sentence to that effect in the “statistical analyses” section of the methods. If that is not the case, then explain how the tests were selected.

Sincerely,

Julian F. Hillyer

Guest Editor

PLOS Pathogens

Kirk Deitsch

Section Editor

PLOS Pathogens

Kasturi Haldar

Editor-in-Chief

PLOS Pathogens

orcid.org/0000-0001-5065-158X

Michael Malim

Editor-in-Chief

PLOS Pathogens

orcid.org/0000-0002-7699-2064

Dear Dr. Povelones,

Your manuscript was evaluated by three reviewers and me. The clear consensus is that this manuscript is a significant scientific contribution because of (i) its mechanistic discoveries on the melanization pathway of a mosquito that is a major vector of malaria in Africa, and (ii) its presentation of a novel, non-invasive method for measuring phenoloxidase activity in mosquitoes.

When revising your manuscript, please pay attention to the reviewer comments.

Reviewer 1 requests elaboration on the CLIPC9 RNAi survival phenotype. The manuscript already draws parallels to what has been published for CLIPA8, but elaboration on this point is warranted. Reviewer 1 also recommends an experiment that you should keep in mind when conducting follow-up studies (so, not required for this submission).

Reviewer 2 presents some technical questions on spontaneous melanization and several Western blots, plus the strength of the model. I believe all of these issues can be addressed in the revised manuscript and rebuttal letter (more on this below).

Reviewer 3 asks for further explanation or clarification on some aspects of the interpretation of the data. Note that reviewers 2 and 3 shared some concern about the interpretation of CLIPC9 cleavage, based on the p12/p30 Western blots. Clarifying this point will be essential in your revision.

In addition, I have several minor comments. On more than one occasion the manuscripts extrapolates the E. coli findings to all Gram(-) bacteria (for example, lines 181-182, 193). This claim is not warranted and should be deleted. To make this claim, the study would have to examine several Gram(-) bacteria and obtain similar findings for all of them. I also recommend that you not use the term “master regulator” (as in the submission letter) or “master negative regulator” (line 330-331) because master regulators are classically defined as transcription factors related to cell differentiation. Finally, multiple statistical tests were used, and they are clearly detailed in the figure legends. I imagine that they were selected after testing the data for normality. If that is the case, include a sentence to that effect in the “statistical analyses” section of the methods. If that is not the case, then explain how the tests were selected.

Sincerely,

Julian Hillyer, PLOS Pathogens Guest Editor

Reviewer Comments (if any, and for reference):

Reviewer's Responses to Questions

**Part I - Summary**

Reviewer #1: This manuscript described the development of new methodology for the study of mosquito factors involved in melanization, and the use of this method to functionally characterize CLIPC9 and other factors in the mosquito's melanization responses to bacteria and Plasmodium. The study involves complex mechanistic analyses that generate new knowledge on multiple factors of the mosquito's complement and melanization-based responses to infection and possibly also injury.

The provided model provided in figure 10D is somewhat limited and could be expanded with he additional factors that are being addressed in this study to provide a more complete picture to the reader: basically an effort should be made din integrating the models provided in the various figures.

Reviewer #2: Sousa et al. report in this manuscript the involvement of Anopheles gambiae CLIPC9 in melanization, a mosquito immune response against malaria parasites, microbial pathogens, and sometimes spontaneously. While mechanism is unclear, a non-destructive method was devised and shown to be a good substitute for the phenoloxidase assay, which is challenging for small insects with a limited amount of hemolymph. Combining the new technique with RNAi screening of the CLIPC genes, alone or in conjunction with one of the CLIPA, TEP1 and other genes, they identified CLIPC9 as the first clip-domain serine protease in group C participating in melanization and discovered its relationships with TEP1, SPCLIP1, CLIPA8, CLIPA28, and SRPN2. These connections are indirect but built on previous studies in this and other insects. The authors also found CLIPA8 is needed for CLIPC9 localization on E. coli bioparticles. The experiments were well deigned and the results were properly interpreted along with inspiring discussions. While the significance and quality of this research have met the standards of PLoS Pathogens, I suggest the authors to improve their manuscript by addressing several questions.

Reviewer #3: The manuscript by Sousa et al is a well-executed study of mosquito melanization, describing an innovative new method to examine melanization using the melanization-associated spot assay, and in utilizing this assay to implicate CLIPC9 in the melanization pathway of An. gambiae. The manuscript is well-written and the results are straight-forward, thorough, and well-presented. Overall, I think it is a great study, and have only minor comments that can be easily corrected in a revised manuscript.

**Part II – Major Issues: Key Experiments Required for Acceptance**

Reviewer #1: (No Response)

Reviewer #2: In the Abstract, the authors stated that their method is based on quantitation of infection-induced excreta. In Fig. 9A and Fig. S7, spontaneous melanization also generate brown spots. How can I be sure if RNAi induced change in melanization is not associated with spontaneous melanization? A systematic check and presentation of additional controls as supplemental materials may be needed to address this concern.

In Fig. 6B (Fig. S8 and Fig. S9), Standard exposure, is CLIPC9-p12 cropped from the image of extended exposure by mistake? Otherwise, I couldn’t explain why there is no major increase in the 12 kDa band intensity, which is apparent for CLIPC9-p30. Note the intensity ratio of the two bands are high on the right. Based on the ratio, I think the authors need to indicate in the figure legends to Fig. 8A-C, Fig. S8, and Fig. S9 whether or not the strips CLIPC9-p12 came from prolonged exposure. How long were the standard and extended exposures?

Reviewer #3: There is nothing that I consider to be a major issue. No further experiments needed.

**Part III – Minor Issues: Editorial and Data Presentation Modifications**

Reviewer #1: The lack of mortality and bacterial melanization in CLIPC9 mosquitoes should be explained and elaborated on. Is the CLIPC9 -regulated defense reaction a minor components of the defense or a post-killing step.

The injection-site melanization of dsCTL-4 treated mosquitoes is interesting and the authors hypothesize hat it could be a result of local infection. That could have been tested using antibiotic treatment.

Reviewer #2: TEP1 cleavage is mediated by a specific proteolytic enzyme in plasma, not necessarily but likely a serine protease. As such, is it theoretically correct to place LRIM1-APL1C-TEP1 on top of the serine protease-serine protease homolog system? I suggest the authors to tune down the current model by emphasizing its tentativeness. At least, not describing TEP1 as “the most upstream one” of the mosquito melanization cascade. Thinking more about the model’s limitations may help to open up ways to overcome them.

I am a little confused by Fig. 5. Are they discussing about “tissue melanization” or “immune melanization” here and the rest of the paper? I understand the definitions were made by other people, not perfectly clear but useful. I would like to see how authors of this manuscript consider their results in relation to the two concepts.

Reviewer #3: -Regarding the results of Figure 5, some additional explanation is needed in the results section to explain differences in melanization and oocyst phenotypes for the double-silencing experiments for CLIPC9/CTL4 and CLIPA8/CTL4. While both CLIPC9 and CLIPA8 are able to negate the influence of CTL4 on melanization, it is unclear why when live oocysts are examined, only CLIPA8-silencing is able to negate the influence of CTL4-silencing on oocyst numbers. The reason for this is unclear.

-Some additional speculation would be appreciated regarding the absence of CLIPC9 cleavage and the production of the p12 product as described in Figure 8. If the p12 band is indicative of an active product that is dependent on upstream CLIPs, could the initial processing of the p30 product be independent of these upstream CLIPs?

-The paragraph containing line 431-454 is interesting and thought provoking. An additional point to consider is that the late-phase immune response where multiple PPOs have been implicated does not involve melanization. Based on Dudzic et al., this might suggest that the effects of CLIPC9 are comparable to Hayan responsible for the blackening reaction (as you describe), leaving the anti-microbial effects of PPOs to potentially be processed by another CLIP.

Specific comments

Line 60: Should be corrected to “vector-borne”

PLOS authors have the option to publish the peer review history of their article (what does this mean?). If published, this will include your full peer review and any attached files.

Reviewer #1: No

Reviewer #2: No

Reviewer #3: No
---

## [Editor Report · Decision Letter 1]

16 Sep 2020

Dear Dr. Povelones,

We are pleased to inform you that your manuscript 'The CLIP-domain serine protease CLIPC9 regulates melanization downstream of SPCLIP1, CLIPA8, and CLIPA28 in the malaria vector Anopheles gambiae' has been provisionally accepted for publication in PLOS Pathogens.

Best regards,

Julian F. Hillyer

Guest Editor

PLOS Pathogens

Kirk Deitsch

Section Editor

PLOS Pathogens

Kasturi Haldar

Editor-in-Chief

PLOS Pathogens

orcid.org/0000-0001-5065-158X

Michael Malim

Editor-in-Chief

PLOS Pathogens

orcid.org/0000-0002-7699-2064

---

## [Editor Report · Acceptance letter]

5 Oct 2020

Dear Dr. Povelones,

We are delighted to inform you that your manuscript, "The CLIP-domain serine protease CLIPC9 regulates melanization downstream of SPCLIP1, CLIPA8, and CLIPA28 in the malaria vector *Anopheles gambiae*," has been formally accepted for publication in PLOS Pathogens.

Best regards,

Kasturi Haldar

Editor-in-Chief

PLOS Pathogens

orcid.org/0000-0001-5065-158X

Michael Malim

Editor-in-Chief

PLOS Pathogens

orcid.org/0000-0002-7699-2064